# Optimization Can Learn Johnson Lindenstrauss Embeddings

**Nikos Tsikouras**
UOA & Archimedes / Athena RC
n.tsikouras@athenarc.gr

**Constantine Caramanis**
UT Austin & Archimedes / Athena RC
constantine@utexas.edu

**Christos Tzamos**
UOA & Archimedes / Athena RC
christos@tzamos.com

## Abstract

Embeddings play a pivotal role across various disciplines, offering compact representations of complex data structures. Randomized methods like Johnson-Lindenstrauss (JL) provide state-of-the-art and essentially unimprovable theoretical guarantees for achieving such representations. These guarantees are worst-case and in particular, neither the analysis, *nor the algorithm*, takes into account any potential structural information of the data. The natural question is: must we randomize? Could we instead use an optimization-based approach, working directly with the data? A first answer is no: as we show, the distance-preserving objective of JL has a non-convex landscape over the space of projection matrices, with many bad stationary points. But this is not the final answer.

We present a novel method motivated by diffusion models, that circumvents this fundamental challenge: rather than performing optimization directly over the space of projection matrices, we use optimization over the larger space of *random solution samplers*, gradually reducing the variance of the sampler. We show that by moving through this larger space, our objective converges to a deterministic (zero variance) solution, avoiding bad stationary points.

This method can also be seen as an optimization-based derandomization approach and is an idea and method that we believe can be applied to many other problems.

## 1 Introduction

Embeddings are foundational across diverse disciplines, offering compact representations of complex data structures. Algorithms across different domains leverage embeddings to capture nuanced relationships between data points, improving efficiency and effectiveness in processing. Within this field, there are distinct lines of work.

Embeddings have been used as dimensionality-reducing tools, while preserving important structures in data. They have been a major research focus for years, and have been a key ingredient in several algorithmic applications such as graph sparsification [Spielman and Srivastava, 2008], nearest-neighbor search [Indyk and Motwani, 1998], hashing [Dietzfelbinger et al., 1997] and digital research [Schmidt, 2018]. A celebrated result in this area has been the Johnson-Lindenstrauss (JL) lemma Johnson [1984] which shows that a random linear mapping can reduce the dimension of the dataset while approximately preserving the $L_2$ norm of all points, with high probability. JL embeddings give bounds on the *maximum distortion* over all the points. Many variants have been developed and studied, as we discuss below in Section 2. For this paper, the salient point is that the algorithms that

38th Conference on Neural Information Processing Systems (NeurIPS 2024).

enjoy theoretical guarantees are for random projections drawn from a distribution over projection matrices. To the best of our knowledge, even among derandomization of JL-type techniques (e.g., Clarkson and Woodruff [2009], and see Section 2 for further discussion), there are no guarantees for optimization-based algorithms that directly attempt to minimize a distortion objective.

On the other hand, in the last decade, embeddings have risen to prominence across various tasks in deep learning. In encoder-decoder architectures, embeddings act as intermediary representations to solve a broad range of challenges in natural language processing and speech processing, [Bengio and Heigold, 2014, Cho et al., 2014a,b, Qi et al., 2018, Rush et al., 2015, Xiong et al., 2016]. In the domain of face recognition, embeddings encode facial features into concise vectors, enabling precise identification and matching [Liu et al., 2015, 2017, Schroff et al., 2015]. Furthermore, contrastive learning techniques utilize embeddings to emphasize disparities between similar and dissimilar instances, thereby augmenting the discriminative capabilities of models [Gao et al., 2021, Khosla et al., 2020, Radford et al., 2021].

In contrast to the JL-type results, in the above applications these embeddings are learned using optimization as part of the (pre)training process [Caron et al., 2021, Oquab et al., 2023, Press and Wolf, 2016, Vaswani et al., 2017]. Though their empirical success is remarkable, the non-convex landscape of the optimization process makes obtaining theoretical guarantees a key challenge.

Another important related area is the extensive body of literature dedicated to employing optimization techniques for Principal Component Analysis (PCA). PCA aims to find linear embeddings that capture the optimal *average distortion* in the data, thereby reducing its dimensionality while preserving as much information as possible. The last decade saw significant success of direct matrix optimization in various PCA-like settings [Garber and Hazan, 2015, Shamir, 2016, Xu and Li, 2021, Yi et al., 2016]. Though clearly there are similarities to the JL objective, the difference between guarantees on the maximum perturbation (JL) vs the average perturbation (PCA) are significant, and one reason why the techniques pioneered for PCA have yet to be applied successfully to JL.

Nevertheless, the empirical success of optimization in neural networks, and its theoretical success for PCA-type objectives, motivate us to revisit the JL low-distortion embedding task (see Section 3 for the exact definition of the *JL guarantee*), and ask the question that to the best of our knowledge, has yet to find an answer:

> *Can optimization-based approaches be used to obtain a matrix*
> *that satisfies the Johnson-Lindenstrauss guarantee?*

Answering this question is the main goal of this paper.

**Our Contributions and a Conceptual Road Map.**

The main contribution of this paper is in developing a framework that allows a direct (and deterministic) optimization approach to obtain the same JL guarantees as random projection. After all, it is well-known that the JL guarantees given by random projection are not improvable [Larsen and Nelson, 2014, 2017, Alon and Klartag, 2017]. Moreover, established derandomization techniques based on conditional expectation and other methods have long been available [Raghavan, 1988, Engebretsen et al., 2002, Bhargava and Kosaraju, 2005].

The key conceptual steps on the way to our main result are as follows:

**Step 1:** We first show that the optimization landscape in the ambient space of projection matrices is not favorable, and in particular, any attempt to directly minimize distortion over this space using first or second-order methods, is destined to fail.

**Theorem** (Informal version of Theorem 1). *The maximum distortion objective considered as a function in the space of matrices has many suboptimal local minima.*

**Step 2:** Given the above negative result, a different approach is required. We draw inspiration from diffusion models and solution samplers [Bello et al., 2016, Ho et al., 2020]. Rather than optimize in the space of matrices, we optimize in the larger space of (mean, variance) parameters of Gaussian distributions over embedding matrices. Thanks to the original JL theorem, we know an initial choice of parameters that define a solution sampler whose expected distortion is small: zero mean and unit variance. Akin to diffusion, we then seek to sequentially decrease the magnitude of the variance, without increasing the expected distortion of the sampler. Note that the space of matrices is properly

contained in this space of samplers, as we can identify a specific (deterministic) projection matrix with a Gaussian distribution with that mean, and zero variance. The challenge is to find a path from our initial JL sampler, to a deterministic sampler (a projection matrix) whose maximum distortion is approximately as good as the expected guarantee of the original JL sampler.

We turn this into an optimization problem by creating an objective function in the space of samplers. Our first result demonstrates that if we find a second-order stationary point for this objective function, then we have solved our original problem. Specifically:

**Theorem** (Informal Version of Theorem 2). *For data $x_1, \ldots, x_n \in \mathbb{R}^d$ and target dimension $k$ all second-order stationary points reachable from the origin for the objective function defined in Equation 4 have zero variance and hence correspond to fixed matrices; moreover, these matrices satisfy the JL guarantee.*

**Step 3**: The final step requires proving the tractability of finding a second-order stationary point, i.e., of solving this optimization problem. We do so *using a generic deterministic second-order optimization algorithm* (see Alg. 1). Thus our result shows that our second-order optimization algorithm successively performs reverse-diffusion-like steps, decreasing the variance without deteriorating the quality of the solution sampler, until it has finally arrived at a deterministic solution.

**Theorem** (Informal Version of Theorem 3). *For data $x_1, \ldots, x_n \in \mathbb{R}^d$ and target dimension $k$ running Algorithm 1 using Equation 4 for $\mathrm{poly}(n, k, d)$ steps returns a matrix that satisfies the JL guarantee.*

**Step 4**: Finally, we show through simulations that the qualitative and quantitative results of our theory are borne out.

## 2   Related Work and Alternative Approaches

The JL lemma is a well-studied result studied in the literature, with several simplifications and extensions of the original proof [Dasgupta and Gupta, 2003, Frankl and Maehara, 1988, Kane and Nelson, 2010, Matoušek, 2008]. It has also been shown that the JL lemma is optimal in terms of the target dimension. The authors in [Larsen and Nelson, 2017, Alon and Klartag, 2017] provide a tight lower bound on the target dimension required by the JL lemma, given a specific distortion, for any random linear mapping. In addition to these theoretical insights, there have been various approaches aimed at efficiently constructing random matrices that satisfy the JL guarantee with high probability.

One approach samples each matrix entry independently from a Gaussian distribution [Indyk and Motwani, 1998], while others utilize Rademacher and sparse Rademacher distributions [Arriaga and Vempala, 2006, Achlioptas, 2001]. Moreover, generalized sampling methods have shown that any distribution with zero mean, unit variance, and a subgaussian tail can be used [Matoušek, 2008]. The Fast Johnson–Lindenstrauss Transform employs sparse matrices and randomised Walsh–Hadamard transforms for efficiency [Ailon and Chazelle, 2009]. Additionally, the subsampled randomised Hadamard transform, achieves efficient embeddings by combining subsampling with randomised Hadamard transforms, maintaining a high probability of preserving the distances between points [Ailon and Liberty, 2013].

**Derandomizing Johnson-Lindenstrauss.** Derandomization is a technique for developing deterministic algorithms or algorithms that require fewer random bits, and it has proven to be a powerful theoretical tool [Kabanets, 2002]. There have been numerous efforts to derandomize the JL lemma, with a significant focus on using pseudorandom generators (PRGs) capable of fooling statistical tests [Nisan, 1990]. These constructions aim to achieve reduced seed lengths while satisfying the JL lemma's norm-preserving properties with $\pm\varepsilon$ distortion and probability of failure $\delta$. For example, the $\ell_2$-streaming algorithm achieves a JL family with seed length $O(\log d)$ and with $k = O(1/(\varepsilon^2\delta))$ [Alon et al., 1996]. The authors in [Clarkson and Woodruff, 2009] leveraged the use of scaled random Bernoulli matrices with $\Omega(\log(1/\delta))$-wise independent entries, resulting in a seed length of $O(\log(1/\delta)\log d)$. Additionally, PRGs that $\delta$-fool degree-2 polynomial threshold functions generate JL families with seed lengths of $\mathrm{poly}(1/\delta)\log d$ [Meka and Zuckerman, 2010].

Other approaches have utilized conditional probabilities and pessimistic estimators, introduced in [Raghavan, 1988], to derive deterministic algorithms for JL embeddings [Engebretsen et al., 2002, Bhargava and Kosaraju, 2005].

These works differ from ours in several ways. The Nisan pseudorandom generator uses a few random bits. Additionally, the authors in [Bhargava and Kosaraju, 2005] fully derandomize the Rademacher construction by [Achlioptas, 2001] using a combinatorial algorithm that greedily selects the best matrix entries. Even in such a coordinate-wise fashion, using a gradient-descent (continuous-optimization) approach is challenging, as even then the optimization landscape is bimodal.

Overall, the key difference in philosophy, setting and ultimately results, comes from our focus on optimization: our method is a study in the power of local (first and second-order) optimization methods.

## 3 Preliminaries and Notation

In this section, we introduce essential definitions and notation for our work. We consider without loss of generality unit norm data points $x_1, \ldots, x_n \in \mathbb{R}^d$, which we aim to project into $k$ dimensions while preserving their norms with distortion at most $\varepsilon = O(\sqrt{\log n/k})$. Specifically, we seek matrices that satisfy the *Johnson-Lindenstrauss guarantee*:

**Definition 1** (Johnson-Lindenstrauss guarantee). *The Johnson-Lindenstrauss guarantee (JL guarantee) states that for given dataset $x_1, \ldots, x_n \in \mathbb{R}^d$ and target dimension $k$, the distortion for all points does not exceed $O(\sqrt{\log n/k})$.*

To achieve this we define a linear mapping $f(x) = Ax$, where $A \in \mathbb{R}^{k \times d}$. The JL Lemma guarantees that there exists a random linear mapping that achieves this projection with high probability:

**Lemma 1** (Distributional Johnson-Lindenstrauss Lemma). *For $\varepsilon, \delta \in (0,1)$ and $k = O(\log(1/\delta)/\varepsilon^2)$, there exists a probability distribution $D$ over linear functions $f : \mathbb{R}^d \to \mathbb{R}^k$ such that for every $x \in \mathbb{R}^d$:*

$$\Pr_{f \sim D} \left( \|f(x)\|_2^2 \in \left[ (1-\varepsilon)\|x\|_2^2, (1+\varepsilon)\|x\|_2^2 \right] \right) \geq 1 - \delta.$$

There has been significant research aimed at improving the construction of these random mappings. In contrast to traditional algorithms, our approach proposes learning the linear mapping directly from the data, leveraging the inherent structure to surpass worst-case performance.

Next, we give essential definitions for our optimization framework.

**Definition 2.** *A function $f : \mathbb{R}^d \to \mathbb{R}$ is defined to be L-smooth, if for all $x, y \in \mathbb{R}^d$ it satisfies:*

$$\|\nabla f(x) - \nabla f(y)\|_2 \leq L\|x - y\|_2.$$

*A function is called $K-$Hessian Lipschitz if for all $x, y \in \mathbb{R}^d$:*

$$\|\nabla^2 f(x) - \nabla^2 f(y)\|_2 \leq K\|x - y\|_2.$$

Below, we give the definition for approximate stationarity.

**Definition 3.** *(Approximate stationarity). For a $K-$Hessian Lipschitz function $f(\cdot)$, we say that a point $x^*$ is a $\rho-$second-order stationary point ($\rho$-SOSP) if:*

$$\|\nabla f(x^*)\|_2 \leq \rho \quad and \quad \lambda_{\min}(\nabla^2 f(x^*)) \geq -\sqrt{K\rho}.$$

**Notation.** For vectors $u, v$ we use $\langle u, v \rangle$ to denote their inner product and $\|u\|_2$ to denote the $L_2$ norm. For matrix $M \in \mathbb{R}^{k \times d}$, we denote the element of the $i^{th}$ row and $j^{th}$ column by $\mu_{i,j}$ and we use $\|M\|_F$ to denote the Frobenius norm. For matrix $M \in \mathbb{R}^{k \times d}$ and $\sigma^2 \in \mathbb{R}^+$ we use $N(M, \sigma^2)$ to denote an $k \times d$ random Gaussian matrix where each element $a_{i,j} \sim N(\mu_{i,j}, \sigma^2)$. We use $\nabla f$ and $\nabla^2 f$ to denote the gradient and Hessian operators, respectively.

# 4 The Main Results

This section contains the full statement of our main theorems, and proof outlines. We organize the flow of this section according to our conceptual outline given in the introduction. In most cases, we defer the full proofs to the appendix.

Our goal is to find a matrix that satisfies the *Johnson-Lindenstrauss guarantee* as given in Lemma 1. Consider the natural objective function of maximum distortion:

$$h(A) = \max_{x_1,\ldots,x_n} \left| \|Ax\|_2^2 - 1 \right|. \tag{1}$$

**Step 1**: The first step tells us what will not work. In particular, direct optimization over the space of matrices cannot work. Our first result shows that minimizing this maximum distortion objective via a first or second-order method, is a doomed approach. In particular, we show that there exist instances which are bad local minima.

**Theorem 1.** *For all $k > 1$, there exists a family of matrices $A^{k \times k+1}$ which are strict local minima for the objective function of Equation 1 reachable from the origin. The achieved distortion is $\Omega(1)$ over a set of $O(k^2)$ points, while there exist matrices yielding distortion $O(\sqrt{\log k/k}) \to 0$.*

The proof of this is constructive. We construct a dataset and then show that a set of matrices reachable from the origin have large constant distortion, yet these points are locally unimprovable. The full proof can be found in Appendix A.1 $\qquad\square$

**Step 2**: The key idea towards our final result is to perform an optimization over an extended space: the space of parameters of random Gaussian solution samplers. We first define an optimization objective over this space, and then prove properties about the resulting landscape over the space of samplers.

A Gaussian solution sampler is defined by its mean and variance. We only consider the case where all entries have the same variance. Thus, our new parameter space consists of pairs $(\boldsymbol{M}, \sigma^2)$, where $\boldsymbol{M}$ is a projection matrix, here interpreted as the mean of a Gaussian distribution, and $\sigma^2$ is the variance parameter. Given $(\boldsymbol{M}, \sigma^2)$, the solution sampler defined is simply: $A \sim N(\boldsymbol{M}, \sigma^2)$.

We note that our new parameter space has just one additional parameter than the ambient setting.

**Step 2A**: We next must extend the maximum distortion objective above, to the space of random solution samplers we have defined. Consider the objective $f^*$, defined as follows:

$$f^*(\boldsymbol{M}, \sigma^2) = \Pr_{A \sim N(\boldsymbol{M}, \sigma^2)} \left[ h(A) > \varepsilon \right]. \tag{2}$$

Thus, $f^*(\boldsymbol{M}, \sigma^2)$ is the probability that a matrix sampled according to the corresponding Gaussian distribution will have maximum distortion at least $\varepsilon$. When we take $\varepsilon = O(\sqrt{\log n/k})$, our objective function $f^*$ gives us the probability that a Gaussian solution sampler fails to produce a matrix that meets the JL guarantee. Hence, a good sampler is one that makes $f^*$ small.

We note that in the proof of the JL lemma in [Indyk and Motwani, 1998], the authors show that a matrix with Gaussian entries satisfies the JL guarantee with high probability. Thus, in particular, we know that taking $\boldsymbol{M} = \boldsymbol{0}$, where $\boldsymbol{0}$ is a $k \times d$ zero matrix, and $\sigma^2 = 1$, gives a low objective value for $f^*$.

In the context of these definitions, therefore, our goal is to find a matrix $\hat{\boldsymbol{M}}$ such that $f^*(\hat{\boldsymbol{M}}, 0)$ has a low objective value. To do this, we now define a related objective value, which thanks to a regularization term, will allow the optimization algorithm to push us towards lower variance solutions. The technical challenge is then to show that we can control any deterioration of the JL guarantee of these lower variance solutions.

We define our final objective function starting from $f^*$ defined above. First, we simplify the objective by applying a standard union bound and write a relaxed objective that sums for every point the probability that the point will have a norm outside the required bounds after the linear transformation.

$$f(\boldsymbol{M}, \sigma^2) = \sum_{j=1}^{n} \mathrm{Pr}_{A \sim N(\boldsymbol{M}, \sigma^2)} \left[ \frac{1}{k} \|A x_j\|_2^2 \notin (1 - \varepsilon, 1 + \varepsilon) \right]. \tag{3}$$

For an appropriately chosen value of $\varepsilon = O(\sqrt{\log n / k})$, we have that no constraint is violated with probability greater than $1/(3n)$ and $f(\boldsymbol{0}, 1) < 1/3$. The function $f$ serves as an upper bound on the probability of generating a matrix that does not have the JL guarantee, effectively acting as a proxy for "bad" events. Next, we add an appropriate regularization term that penalizes high variance points. Our overall objective is thus:

$$g(\boldsymbol{M}, \sigma^2) = f(\boldsymbol{M}, \sigma^2) + \sigma^2/2. \tag{4}$$

At our initialization point $\boldsymbol{M} = \boldsymbol{0}$ and $\sigma^2 = 1$, the value of the regularization is $1/2$, thus: $g(\boldsymbol{0}, 1) < 1/3 + 1/2 < 5/6$. This is crucial, as it implies that following any decreasing path in $g$ leads to points with a likelihood of a bad event being less than 1. Consequently, this convergence toward a solution sampler maintains a positive (and $O(1)$) probability of achieving a projection matrix that satisfies the JL guarantee. The next step provides our algorithm. After that, we characterize its fixed points.

**Step 2B**: Algorithm 1 is a second-order descent algorithm consisting of two simple steps: At a given point $x_t = (\boldsymbol{M}_t, \sigma_t)$, if the gradient is sufficiently large, we take a gradient step. If the gradient is small, we consider the Hessian; if the smallest eigenvalue is sufficiently negative, we take a step in that direction of negative curvature; otherwise the algorithm terminates by reporting *the mean parameter $\boldsymbol{M}_t$* (see Lemma 4 for discussion on this final point). To prove correctness of the algorithm, we must show that any $\rho$-SOSP will have sufficiently small variance. The proof of correctness is given in **Step 2C**, and a bound on its running time in **Step 3**. We can call this algorithm recursively, to find the best distortion, using a simple routine given in Algorithm 2.

We note that in principle, many first-order methods can also be used, for example, Perturbed Gradient Descent (PGD) which has been shown to converge to second-order stationary points fast [Jin et al., 2017]. We use a *deterministic algorithm* in our analysis to enable a straightforward derandomization of the JL lemma through the optimization of Equation 4.

---

**Algorithm 1** Hessian Descent.

---

**Require:** $\nabla g, \nabla^2 g, \nu = \frac{1}{L}, h = \frac{3\sqrt{\rho}}{K}, L, K, \rho, \boldsymbol{M}_{\mathrm{init}} = \boldsymbol{0}, \sigma_{\mathrm{init}}^2 = 1$
1: $t \leftarrow 0$
2: $x_t \leftarrow (\boldsymbol{M}_{\mathrm{init}}, \sigma_{\mathrm{init}}^2)$
3: **while** true **do**
4:     **if** $\|\nabla g(x_t)\| > \rho$ **then**
5:         $x_{t+1} \leftarrow x_t - \nu \cdot \nabla g(x_t)$
6:     **else if** $\|\nabla g(x_t)\|_2 \leq \rho$ and $\lambda_{\min}(\nabla^2 g(x_t)) < -\sqrt{K\rho}$ **then**
7:         $u_1 \leftarrow$ the eigenvector corresponding to $\lambda_{\min}(\nabla^2 g(x_t))$
8:         $x_{t+1} \leftarrow x_t + h u_1$
9:     **else**
10:         **return** $x_t[0] = M_t$
11:     **end if**
12:     $t \leftarrow t + 1$
13: **end while**

---

**Step 2C**: Since we initialize at a good solution sampler and we use a descent algorithm on our objective, we know that we can never move to a bad sampler. But that is not enough for us. For we recall that our goal is not to find a good randomized algorithm, but rather to find a good (deterministic) JL matrix, via optimization. Thus we must show that we do not get trapped in any points that have non-zero variance.

We accomplish this in several lemmas. First in Lemma 2 we show that stationary points must have zero variance. We then refine this in Lemma 3 and show that being in a $\rho$-second-order stationary point requires the variance to be very small. We need this in order to show we can escape from any point with sufficiently large variance. Finally, in Theorem 2 we show that our second-order algorithm

**Algorithm 2** An algorithm to find optimal distortion.

---
**Require:** $\nabla g, \nabla^2 g, L, K, \rho, x_{\text{initial}}, \varepsilon_{\text{grid}}$
 1: $\min_\varepsilon \leftarrow \infty$
 2: $\min_{\text{value}} \leftarrow \infty$
 3: **for** each $\varepsilon$ in $\varepsilon_{\text{grid}}$ **do**
 4:     $M_\varepsilon \leftarrow$ HESSIAN DESCENT$(\nabla g, \nabla^2 g, \nu, h, L, K, \rho)$
 5:     value $\leftarrow$ max distortion of $M_\varepsilon$
 6:     **if** value $< \min_{\text{value}}$ **then**
 7:         $\min_{\text{value}} \leftarrow$ value
 8:         $\min_\varepsilon \leftarrow \varepsilon$
 9:     **end if**
10: **end for**
11: **return** $\min_\varepsilon, \min_{\text{value}}$

---

will not get stuck at any point with large variance, and that once we are at a solution sampler with small enough variance, the mean parameter itself will enjoy (deterministically) the JL guarantee.

In the following lemma, we show that points with non-zero variance cannot be local minima. Specifically, we show that for any given mean matrix and variance, there exists a nearby mean matrix with reduced variance that improves the objective value.

**Lemma 2.** *Let $M \in \mathbb{R}^{k \times d}$, and $\sigma > 0$. Then for any $\gamma \in [0, \sigma]$, there exists $M' \in \mathbb{R}^{k \times d}$ such that:*

- $\|M - M'\|_F \leq 2\gamma\sqrt{kd \log\left(\frac{3\sqrt{kd}}{\gamma}\right)}$,

- $g(M', \sigma^2 - \gamma^2) \leq g(M, \sigma^2) - \gamma^2/6$.

**Proof Sketch:** The proof of the lemma essentially is a small derandomization step, where we show that by taking a sufficiently small variance-reducing step, even if we deteriorate the JL guarantee (i.e., the function $f$ increases), the decrease in the regularizer outweighs this increase, thereby decreasing the overall value of the objective, thus showing we could not have been at a local minimum.

More specifically, the proof proceeds as follows. We begin with a Gaussian matrix $A \sim N(M, \sigma^2)$ and use the additivity property to partition it: $A = A^\gamma + A'$, where $A^\gamma \sim N(0, \gamma^2)$ and $A' \sim N(M, \sigma^2 - \gamma^2)$, representing small additive noise and the remainder of $A$, respectively.

The core idea is to derandomize $A^\gamma$ to achieve a decreased objective value. We extend the definition of a bad event by constraining $A^\gamma$ to take values only within a specific range $R$. Within $R$, there must exist a realization of $A^\gamma$, denoted as $\alpha^\gamma$, which at worst, only slightly increases the probability of failure due to the truncation of the Gaussian distribution tails. By choosing this specific value $\alpha^\gamma$, we effectively derandomize $A^\gamma$. As we show in the appendix, this can result in an only slightly increased probability of a bad event.

We then define $M' = \alpha^\gamma + M$ and show that the regularization term ensures an overall decrease in the objective function. The full proof can be found in Appendix A.2. $\qquad\square$

In Lemma 2, we established that any point with non-zero variance cannot be a local minimum, as there always exists a nearby point with lower objective value. The next lemma addresses whether a descent direction can be found at each step. We prove that this is indeed the case, specifically demonstrating that the $\rho$-second-order stationary points of the objective function in Equation 4 correspond to points with approximately zero variance.

**Lemma 3.** *Consider $x_1, \ldots, x_n \in \mathbb{R}^d$. Given target dimension $k$ choose $\varepsilon = O(\sqrt{\log n/k})$. The $\rho$-second-order stationary points of the objective function in Eq. 4 implies $\sigma^2 < \text{poly}(n, k, d) \cdot \rho^{O(1)}$.*

**Proof Sketch:** We establish the lemma by examining the behavior of the variance $\sigma^2$ at points approaching $\rho$-second-order stationarity under the objective function defined in Equation 4. While $\sigma^2$ is large we employ a series of incremental reductions, and we show we can continue in this manner until $\sigma^2$ is reduced at least to the claimed level.

Using Taylor's theorem and the Lipschitzness of $\nabla^2 g$, we prove that at any point $(\boldsymbol{M}, \sigma^2)$, either the gradient will be large and thus progress will be made using first-order methods, or that the minimum eigenvalue of the Hessian will be negative and thus we can follow that direction to make progress. We then show that convergence to $\rho$-second-order stationary points gives us the desired result. Controlling the effect of the Lipschitz constant is a main challenge. The full proof can be found in Appendix A.3. $\qquad\square$

**Step 2D**: Since our Algorithm 1 finds an approximate $\rho$-SOSP, we need an additional result that gives us a stopping criterion once the variance is small enough, and simply use the mean with controlled deterioration of the JL guarantee. That is, instead of sampling from $A \sim N(\boldsymbol{M}, \sigma^2)$, we can directly use $\boldsymbol{M}$ instead. This is why in line 10 of Algorithm 1, we simply return the parameter $\boldsymbol{M}_t$.

**Lemma 4.** *Given $n$ unit vectors in $\mathbb{R}^d$ and a target dimension $k$, choose $\varepsilon = O(\sqrt{\log n/k})$ such that distribution $A \sim N(\boldsymbol{M}, \sigma^2)$ satisfies the JL guarantee with distortion $\varepsilon$ with probability $1/6$. Then using matrix $\boldsymbol{M}$ instead of sampling from $A$ retains the JL guarantee with a threshold increased by at most $\mathrm{poly}(\sigma, 1/k)$.*

**Proof Sketch:** By assumption, $A \sim N(\boldsymbol{M}, \sigma^2)$ satisfies $1/k\|Ax\|_2^2 \in (1 - \varepsilon, 1 + \varepsilon)$ with probability at least $1/6$. Next, we decompose $A$ as $A = \boldsymbol{M} + Z$ with $Z \sim N(\boldsymbol{0}, \sigma^2)$, and invoking the JL lemma, we choose $\varepsilon_0$ such that $1/k\|Zx\|_2^2 \in [\sigma^2(1 - \varepsilon_0), \sigma^2(1 + \varepsilon_0)]$ with probability at least $6/7$. This choice ensures that there exists a region in the space such that the JL guarantee holds simultaneously for both $A$ and $Z$. Our goal is to establish a bound on the distortion when using $\boldsymbol{M}$ instead of sampling from $A$. We show that $\|\boldsymbol{M}x\|_2^2$ can be bounded with terms involving only $\|Ax\|_2^2$ and $\|Zx\|_2^2$ and derive an upper and lower bound on the distortion that must hold deterministically. The full proof can be found in Appendix A.4. $\qquad\square$

Putting the above results together, we have our first main result that shows the correctness of Algorithm 1: it will not get trapped at any point with a large variance, and once it does finally arrive at a point of small enough variance, the mean parameter satisfies the JL property.

**Theorem 2.** *Given $n$ unit vectors in $\mathbb{R}^d$ and a target dimension $k$, consider the corresponding function $g$ of Equation 4 for any $\varepsilon \geq C\sqrt{\log n/k}$ where $C$ is a sufficiently large constant. Any $\rho$-SOSP, that is a pair of parameters $(\boldsymbol{M}, \sigma^2)$, of $g$ reachable from the origin satisfies $\sigma^2 < \mathrm{poly}(n, k, d) \cdot \rho^{O(1)}$ and goes to 0 as $\rho \to 0$. Moreover, when $\rho < 1/\mathrm{poly}(n, k, d)$, $\boldsymbol{M}$ satisfies the JL guarantee having distortion at most $O(\varepsilon)$.*

*Proof.* We choose the parameter $C$ such that $g(\boldsymbol{0}, 1) < 5/6$. Using Lemmas 2 and 3 we find that a $\rho$-SOSP of $g$ is a pair of parameters $(\boldsymbol{M}, \sigma^2)$, with $\sigma^2 < \mathrm{poly}(n, k, d) \cdot \rho^{O(1)}$ and $g(\boldsymbol{M}, \sigma^2) < g(\boldsymbol{0}, 1)$. This implies that drawing sample from $A \sim N(\boldsymbol{M}, \sigma^2)$ satisfies the JL guarantee with distortion at most $\varepsilon$ with probability $1/6$.

Then, from Lemma 4, using the matrix $\boldsymbol{M}$ satisfies the JL guarantee with distortion $O(\epsilon)$. $\qquad\square$

**Step 3**: The above result proves correctness of Algorithm 1, but is qualitative: we have not proved how many steps are required. Below we give a quantitative result proving that we can minimize our objective function efficiently and learn a deterministic JL embedding in polynomial time, while incurring a minor increase in the distortion. Though as mentioned, we see the main contribution of our work as centering on the optimization formulation, we note that this theorem constitutes a novel approach to derandomizing the Gaussian JL transformation.

**Theorem 3.** *Given $n$ unit vectors in $\mathbb{R}^d$ and a target dimension $k$, consider any $\varepsilon \geq C\sqrt{\log n/k}$ where $C$ is a sufficiently large constant. Then, running Algorithm 1 to deterministically optimize the objective function $g$ of Equation 4 for $\mathrm{poly}(n, k, d)$ steps returns a matrix $\boldsymbol{M}$ that satisfies the JL guarantee with distortion at most $O(\varepsilon)$.*

*Proof.* The first part of this theorem relies on two auxiliary results:

**Lemma 5** (Sufficient Descent for Gradient Descent). *If $\|\nabla g(x_t)\|_2 > \rho$ and $L$ the smoothness of $g$, then for $\nu = \frac{1}{L}$ and $x_{t+1} = x_t - \nu \cdot \nabla g(x_t)$, we have $g(x_{t+1}) \leq g(x_t) - \frac{\nu\rho^2}{2}$.*

**Lemma 6** (Sufficient Descent for Negative Curvature Descent). *If $\|\nabla g(x_t)\|_2 \leq \rho$ and $\lambda_{min}(\nabla^2 g(x_t)) < -\sqrt{K\rho}$, and $K$ the Hessian Lipschitz parameter, then for $h = \frac{3\sqrt{\rho}}{K}$ and $x_{t+1} = x_t + hu_1$, where $u_1$ corresponds to the eigenvector of the minimum eigenvalue, we have $g(x_{t+1}) \leq g(x_t) - \frac{3\rho^{1.5}}{4\sqrt{K}}$.*

The proofs of both lemmas are in Appendices A.5, A.6, respectively. Given these two results, now standard optimization techniques show that Algorithm 1 finds a $\rho$-SOSP in $O(1/\rho^{1.5})$ steps.

According to Theorem 2, choosing $\rho < 1/\text{poly}(n, k, d)$, implies that a $\rho$-SOSP for $g$ is a pair of parameters $(\boldsymbol{M}, \sigma^2)$ with $\boldsymbol{M}$ satisfying the JL guarantee with distortion at most $O(\varepsilon)$. Therefore, running Algorithm 1 for $\text{poly}(n, k, d)$ steps returns a matrix $\boldsymbol{M}$ which satisfies the JL guarantee with distortion at most $O(\varepsilon)$. $\qquad\square$

## 5 Simulations

We empirically validate our theoretical results, demonstrating that first and second-order information suffices to learn high-quality embeddings. Our goal is to identify an optimal matrix $\boldsymbol{M}$ that produces embeddings satisfying the JL guarantee (see Definition 1) while minimizing distortion. We show that our method generates embeddings with significantly lower distortion compared to those obtained using the Gaussian construction of the JL lemma.

We generate a unit norm dataset with $n = 100$ data points in $d = 500$ dimensions, and target dimension $k = 30$ dimensions. We aim to minimize the expected maximum distortion $E_{A \sim N(\boldsymbol{M}, \sigma^2)}[h(A)]$ (see Equation 1). We employ the first-order optimization algorithm Adam [Kingma and Ba, 2014] over 5000 iterations and compare our method against the average and minimum distortions over 1000 trials using the baseline matrix $Z \sim N(\boldsymbol{0}, 1)$ (left plot of Figure 1). To calculate the distortions of our method, we sample from the updated mean matrix and variance at each iteration.

As predicted by our theoretical analysis, we demonstrate that while the Gaussian randomized construction achieves satisfactory distortion levels, our method converges to a high-quality deterministic solution that nearly eliminates distortion (both plots of Figure 1). At the conclusion of the procedure, we calculate the distortion using the resultant mean matrix $\boldsymbol{M}$. Interestingly, the results demonstrate that $\|\boldsymbol{M}x\|_2^2 \approx 1$ (i.e. almost 0 distortion), compared to approximately 1 and 0.6 for the average and minimum distortions of the random construction, respectively. We plot the progression of the distortions and variance over the iterations in the plots of Figure 1.

This evidence highlights the effectiveness of our methodology in practice, showcasing the advantage of integrating data structure into dimensionality reduction for more accurate embeddings with provable guarantees.

Further details can be found in Appendix B.

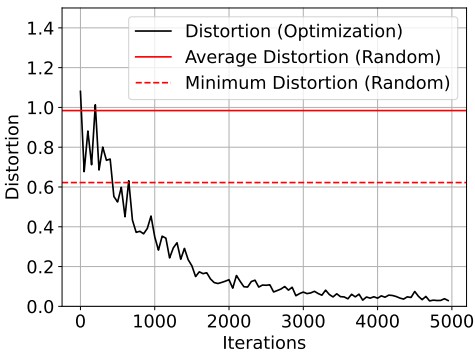 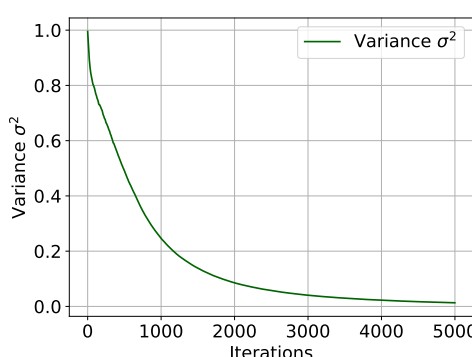

Figure 1: Plot of the distortion obtained through optimization over 5000 iterations vs the average distortion using a random Gaussian matrix (left plot), and the progression of variance over the same number of iterations (right plot). To calculate the distortions' progression with our method, we sample from the updated mean matrix and variance at each iteration and compute the distortion. We remark that the distortion plotted is a proxy for our objective in Equation 4. We observe that our optimization-based approach converges to a deterministic solution sampler. By using the mean matrix $\boldsymbol{M}$, we achieve nearly optimal distortion, where $|\boldsymbol{M}x| \approx |x|$.

# 6    Conclusion

In this work, we have demonstrated that optimization used directly in the sampler space can find a deterministic JL-quality embedding. While our initial focus has been on the relatively simple random construction of the Johnson-Lindenstrauss Lemma, there exist more complex randomized constructions that may offer greater flexibility. We believe that the methods we have introduced could find applicability far outside the JL setting.

## Acknowledgements

This work has been partially supported by project MIS 5154714 of the National Recovery and Resilience Plan Greece 2.0 funded by the European Union under the NextGenerationEU Program.

Constantine Caramanis was partially supported by the NSF IFML Institute (NSF 2019844), and the NSF AI-EDGE Institute (NSF 2112471).

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

# Appendix

# A    Proofs of Section 4

## A.1    Proof of Theorem 1

**Theorem.** *For all $k > 1$, there exists a family of matrices $A^{k \times k+1}$ which are strict local minima for the objective function of Equation 1 reachable from the origin. The achieved distortion is $\Omega(1)$ over a set of $O(k^2)$ points, while there exist matrices yielding distortion $O(\sqrt{\log k/k}) \to 0$.*

*Proof.* We show that trying to minimize the maximum distortion of points directly can lead to getting stuck in bad local minima. Specifically, there exists a set of points such that the family of matrices $\mathcal{F} \subseteq \mathbb{R}^{k \times (k+1)}$, which consist of unitary matrices scaled by 2 in the first $k \times k$ block and a zero vector in the last column, is a local minimum, and any matrix $A'$ with $\|A' - A\|_F^2 < 1/\text{poly}(k)$ for some matrix $A \in \mathcal{F}$ results in worse maximum distortion.

Consider any matrix $A'$ and its closest matrix $A \in \mathcal{F}$. $A'$ can be written as $[A'_{[k \times k]}|v]$ where $A'_{[k \times k]}$ represents the first $k \times k$ block of $A'$ and $v$ its last column. If we compute the polar decomposition of $A'_{[k \times k]} = UP$, where $U$ is a unitary matrix and $P$ is a positive semidefinite symmetric matrix, the closest matrix $A$ to $A'$ can be written as $[2U|0]$. By rotating the space with $U^{-1}$, we assume, without loss of generality, that $U = I$ and thus $A'_{[k \times k]}$ is a positive definite symmetric matrix.

**Dataset Construction:** Define vectors $e_i \in \mathbb{R}^k$ for $i = 1, \ldots, k$. The first $k$ components of our vectors will be either $e_i$ or $e_i + e_j$ with $i \neq j$. For each such vector we have 4 different versions for the last coordinate, denoted by $\text{last}(x)$, is given by:

$$\text{last}(x) \in \{+\sqrt{15}\|x\|_2, -\sqrt{15}\|x\|_2 + \sqrt{7}/3\|x\|_2, -\sqrt{7}/3\|x\|_2\}$$

Thus, the data points are constructed as $\tilde{x} = (x, \text{last}(x))$. Essentially, $\text{last}(x)$ is the completion of the $x$ depending on whether we want tight large distortion or tight small distortion.

**Distortion Analysis:**

We analyze $\|A\tilde{x}\|_2^2 - \|A'\tilde{x}\|_2^2$. Using the Taylor expansion:

$$\|A\tilde{x}\|_2^2 - \|A'\tilde{x}\|_2^2 = 2(A\tilde{x})^T(A - A')\tilde{x} + O(\|A' - A\|_F^2)$$

Since $\|A - A'\|_F^2 = \|P - 2I\|_F^2 + \|v\|_2^2$, let $\|A - A'\|_F^2 = \gamma^2$. We consider two cases based on the magnitude of $\|v\|_2^2$.

**Case 1:** $\|v\|_2^2 \geq \gamma^2/2$

In this case, there exists at least one $i$ such that $v_i^2 \geq \gamma^2/(2k)$. We consider:

$$2(A\tilde{x})^T(A - A')\tilde{x} = x^T(P - 2I)x + x^Tv\,\text{last}(x)$$

Choosing $x = e_i$, we encounter two sub-cases, depending on the sign of $x^T(P - 2I)x$. In both sub-cases, the term $2(A\tilde{x})^T(A - A')\tilde{x}$ dominates the higher-order term. This holds because:

$$|x^T(P - 2I)x + x^Tv\text{last}(x)| > |x^T(P - 2I)x + \sqrt{7}/3\gamma| > \gamma^2 = \|A - A'\|_F^2.$$

Then, we have:

- If $x^T(P - 2I)x < 0$, select the negative value of the tight large distortion case for $\text{last}(x) = -\sqrt{7}/3\|e_i\|$ to ensure $\|A\tilde{x}\|_2^2 < \|A'\tilde{x}\|_2^2$, resulting in worse distortion. That is, we expected norm $4/3$, but instead using matrix $A$, we got norm 2, and any $A'$ nearby will only increase gap.

- If $x^T(P - 2I)x \geq 0$, select the positive value of the tight small distortion case for $\text{last}(x) = \sqrt{15}\|e_i\|_2$ to ensure $\|A\tilde{x}\|^2 > \|A'\tilde{x}\|^2$, again resulting in worse distortion. resulting in worse distortion. In this case, we expected norm 4, but instead we got norm 2, and any $A'$ nearby will only increase gap.

**Case 2:** $\|v\|_2^2 < \gamma^2/2$

This implies that $\|P - 2I\|_F^2 \geq \gamma^2/2$. Again, we encounter two sub-cases. Either an entry on the diagonal of $P - 2I$ has mass $\geq \gamma/2k^2$ or an off-diagonal entry does. We shall prove that in each case, the first-order term dominates.

**-Diagonal Entry:** If $|(P - 2I)_{i,i}| \geq \gamma/2k^2$, choose $x = e_i$. Then:

$$|x^T(P - 2I)x + x^T v\text{last}(x)| > |\gamma/2k^2 + x^T v\text{last}(x)| > \gamma^2 = \|A - A'\|_F^2.$$

We note here that we always match the sign of $\text{last}(x)$ to that of $x^T(P - 2I)x$ to increase the error. Now, we can follow the same logic as in Case 1 to show that there always exists a point with worse distortion.

**-Off-Diagonal Entry:** If $|(P - 2I)_{i,j}| \geq \gamma/2k^2$ for some $i \neq j$, choose $x = e_i + e_j$. Since $P$ is symmetric, we are adding the contribution of two elements, and even if the diagonal elements have the opposite sign, by definition of this sub-case, their magnitude does not suffice to make the norm of the first-order term negligible:

$$|x^T(P - 2I)x + x^T v\text{last}(x)| > |\gamma/2k^2 + x^T v\text{last}(x)| > \gamma^2 = \|A - A'\|_F^2.$$

We can follow the same logic as in Case 1 to show that there always exists a point with worse distortion.

**Conclusion:**

In summary, in all cases, the first-order term $x^T(P - 2I)x + x^T v\text{last}(x)$ dominates over higher-order terms $O(\|A - A'\|_F^2)$, and thus the sign is always determined by the first term. We have shown that there always exists a point $x$ causing increased distortion, thus confirming that any perturbation $\|A' - A\|_F^2 < 1/\text{poly}(k)$ results in worse maximum distortion. Thus, this family of matrices is a local minimum with constant distortion. $\qquad\square$

## A.2 Proof of Lemma 2

**Lemma.** *Let $M \in \mathbb{R}^{k \times d}$, and $\sigma > 0$ and consider a random matrix $A \sim N(M, \sigma^2)$. Then for any $\gamma \in [0, \sigma]$, there exists $M' \in \mathbb{R}^{k \times d}$ such that:*

- $\|M - M'\|_F \leq 2\gamma\sqrt{kd \log\left(\frac{3\sqrt{kd}}{\gamma}\right)}$

- $g(M', \sigma^2 - \gamma^2) \leq g(M, \sigma^2) - \gamma^2/6$

*Proof.* Consider a random matrix $A \sim N(M, \sigma^2)$. Let $\gamma \in [0, \sigma)$, then using the additivity property of Gaussian distributions, we decompose $A$ into $A^\gamma \sim N(0, \gamma^2)$ and $A' \sim N(M, \sigma^2 - \gamma^2)$, such that $A = A^\gamma + A'$. We note that the case for $\gamma^2 = \sigma^2$ follows from the same analysis but taking $A' = M$ deterministically.

We extend the definition of a "bad event", and consider a failure when $\|Ax\|_2$ falls outside the range $R_1 := \left(\sqrt{k(1-\varepsilon)}, \sqrt{k(1+\varepsilon)}\right)$ for any $x$, or when any entry generated from $A^\gamma$ falls beyond the range $R_2 := \left[-2\gamma\sqrt{\log\left(\frac{\sqrt{kd}}{\gamma/3}\right)}, 2\gamma\sqrt{\log\left(\frac{\sqrt{kd}}{\gamma/3}\right)}\right]$. This extension results in an increased probability of a bad event by $\gamma^2/3$. Denote by $a_{i,j}$ the $j^{th}$ element of the $i^{th}$ row of matrix $A^\gamma$, then:

$$\Pr\left(|a_{i,j}| \geq 2\gamma\sqrt{\log\left(\frac{\sqrt{kd}}{\gamma/3}\right)}, \forall i = 1, \ldots, k, \forall i = 1, \ldots, d\right) = \sum_{i=1}^{k}\sum_{j=1}^{d}\left(|a_{1,i}| \geq 2\gamma\sqrt{\log\left(\frac{\sqrt{kd}}{\gamma/3}\right)}\right)$$

$$\leq \sum_{i=1}^{k}\sum_{j=1}^{d} 2\exp\left(-\frac{4\gamma^2\log\left(\frac{\sqrt{kd}}{\gamma/3}\right)}{2\gamma^2}\right)$$

$$= \sum_{i=1}^{k}\sum_{j=1}^{d} 2\exp\left(\log\left(\frac{\gamma^2/9}{kd}\right)\right)$$

$$= \sum_{i=1}^{k}\sum_{j=1}^{d} 2\frac{\gamma^2/9}{kd}$$

$$= 2\gamma^2/9 < \gamma^2/3.$$

We note that the inequality holds due to standard Gaussian properties.

We prove that there exists a realization for $A^\gamma \sim N(\mathbf{0}, \gamma^2)$ within the range $R_2$, which leads to an improvement in the objective function. To see this, we express Equation 3 as follows:

$$f(\mathbf{M}, \sigma^2) = \sum_{j=1}^{n} \Pr_{A \sim \mathcal{N}(\mathbf{M}, \sigma^2)}\left(\|Ax_j\|_2 \notin R_1\right)$$

$$= \sum_{j=1}^{n} \Pr_{A^\gamma, A'}\left[\|A^\gamma x_j + A'x_j\|_2 \notin R_1\right]. \tag{5}$$

Let $\delta$ denote the current value of $f(\mathbf{M}, \sigma^2)$ and consider the following expression of the extended definition of a bad event:

$$\sum_{j=1}^{n} \Pr_{A^\gamma, A'}\left[\|A^\gamma x_j + A'x_j\|_2 \notin R_1 \vee A^\gamma \notin R_2\right]$$

$$= \sum_{j=1}^{n} \Pr_{A^\gamma, A'}\left[\|A^\gamma x_j + A'x_j\|_2 \notin R_1\right] + \Pr_{A^\gamma}\left[A^\gamma \notin R_2\right]$$

$$= E_{A^\gamma}\left(\sum_{j=1}^{n} \Pr_{A'}\left[\|A^\gamma x_j + A'x_j\|_2 \notin R_1 \mid A^\gamma\right] + \mathbf{1}[A^\gamma \notin R_2]\right). \tag{6}$$

We lower bound Equation 6 by the minimum of the function inside the expectation to show that there must exist a realization of $A^\gamma$ that lies in the range $R_2$, and increases the original objective function from $\delta$ to $\delta + \frac{\gamma^2}{3}$ :

$$\min_{A^\gamma \in R_2} \sum_{j=1}^{n} \Pr_{A'}\left[\|A^\gamma x_j + A'x_j\|_2 \notin R_1 \mid A^\gamma\right] < E_{A^\gamma}\left(\sum_{j=1}^{n} \Pr_{A'}\left[\|A^\gamma x_j + A'x_j\|_2 \notin R_1 \mid A^\gamma\right] + \mathbf{1}[A^\gamma \notin R_2]\right)$$

$$< \delta + \gamma^2/3.$$

Denote $\boldsymbol{\alpha}^\gamma := \arg\min_{A^\gamma \in R_2} \sum_{j=1}^{n} \Pr_{A'}\left[\|A^\gamma x_j + A'x_j\|_2 \notin R_1 \mid A^\gamma\right]$, then $A = \boldsymbol{\alpha}^\gamma + A' \sim N(\mathbf{M'}, \sigma^2 - \gamma^2)$, where $\mathbf{M'} = \boldsymbol{\alpha}^\gamma + \mathbf{M}$. Consequently, this means that:

$$f(\mathbf{M'}, \sigma^2 - \gamma^2) = \sum_{j=1}^{n} \Pr_A\left[\|Ax_j\|_2 \notin R_1\right] < \delta + \gamma^2/3.$$

However, since we are working with the regularized objective in Equation 4, reducing $\sigma^2$ to $\sigma^2 - \gamma^2$ means that we have an overall decrease of the objective.

$$g(\boldsymbol{M}', \sigma^2 - \gamma^2) - g(\boldsymbol{M}, \sigma^2) < (\delta + \gamma^2/3 - \gamma^2/2) - \delta < \gamma^2/3 - \gamma^2/2 < -\gamma^2/6.$$

$\square$

### A.3 Proof of Lemma 3

**Lemma.** *Consider $x_1, \ldots, x_n \in \mathbb{R}^d$. Given target dimension $k$ choose $\varepsilon = O(\sqrt{\log n/k})$. The $\rho$-second-order stationary points of the objective function in Equation 4 imply $\sigma^2 < \mathrm{poly}(n, k, d) \cdot \rho^{O(1)}$.*

*Proof.* Assume that we have reached a point $v = (\boldsymbol{M}, \sigma^2)$. Our analysis proceeds in two distinct cases based on the magnitude of $\sigma^2$. If the variance exceeds a specified threshold, we employ a series of incremental reductions. This procedure continues until the variance reaches a sufficiently low threshold, facilitating a transition to a scenario wherein the variance can be reduced to zero in a single step.

**Case 1**: $\sigma^2 > 2^{-5}(Kd^{3/2}\log(kdK)^{3/2})^{-2}$. Choose $\gamma^2 = 2^{-5}(Kd^{3/2}\log(kdK)^{3/2})^{-2}$. According to Lemma 2, there exists a neighboring point $\boldsymbol{M}'$ and we define $v' = (\boldsymbol{M}', \sigma^2 - \gamma^2)$ which reduces the objective function. Then, the distance between $v'$ and $v$ is given by:

$$\|v' - v\|_2 = \sqrt{4\gamma^2 d \log\left(\frac{\sqrt{kd}}{\gamma/3}\right) + \gamma^4}.$$

Denote $K$ the Hessian Lipschitz constant, then, from Lemma 2 and Taylor's theorem on the Lipschitzness of $\nabla^2 g$ [Nesterov, 2013] we get:

$$\left| g(v') - g(v) - \nabla g(v) \cdot (v' - v) - \frac{1}{2}(v' - v) \cdot \nabla^2 g(v) \cdot (v' - v) \right| \leq \frac{K}{6}\|v' - v\|_2^3$$

$$\nabla g(v) \cdot (v' - v) + \frac{1}{2}(v' - v) \cdot \nabla^2 g(v) \cdot (v' - v) - \frac{K}{6}\|v' - v\|_2^3 \leq g(v') - g(v)$$

$$\nabla g(v) \cdot (v' - v) + \frac{1}{2}(v' - v) \cdot \nabla^2 g(v) \cdot (v' - v) - \frac{K}{6}\|v' - v\|_2^3 \leq -\frac{\gamma^2}{6}.$$

This implies that either

$$\nabla g(v) \cdot (v' - v) < -\frac{\gamma^2}{12} \implies \|\nabla g(v)\|_2 > \frac{\gamma}{12\sqrt{4d\log\left(\frac{\sqrt{kd}}{\gamma/3}\right) + \gamma^2}},$$

or

$$(v' - v) \cdot \nabla^2 g(v) \cdot (v' - v) - \frac{K}{6}\|v' - v\|_2^3 < -\frac{\gamma^2}{12}.$$

The last inequality implies that

$$\lambda_{\min}(\nabla^2 g(v)) < -\frac{1}{12\left(4d\log\left(\frac{\sqrt{kd}}{\gamma/3}\right) + \gamma^2\right)} + \frac{K}{6}\sqrt{4\gamma^2 d\log\left(\frac{\sqrt{kd}}{\gamma/3}\right) + \gamma^4}$$

$$< -\frac{1}{24\left(4d\log\left(\frac{\sqrt{kd}}{\gamma/3}\right) + \gamma^2\right)}. \tag{7}$$

The last inequality follows from the choice of $\gamma^2$ we made at the beginning.

This guarantees that in each step either the gradient will be large and thus progress will be made using first-order methods or that the minimum eigenvalue of the Hessian will be negative and thus there exists a direction which we can follow by a second-order method.

**Case 2**: $\sigma^2 \leq 2^{-5}(Kd^{3/2}\log(kdK)^{3/2})^{-2}$.

Assume that after several iterations, we have reached a point $v = (\boldsymbol{M}, \sigma^2)$. Given that the inequality in 7 holds for all $\gamma^2 \leq \sigma^2$, we select the maximum allowable reduction. This allows us to reduce the variance to zero in a single step. Based on Lemma 2, there exists a neighboring point $v' = (\boldsymbol{M}', 0)$ which reduces the objective function and by following the steps as in the previous case, we get:

$$\|\nabla g(v)\|_2 > \frac{\sigma}{12\sqrt{4d\log\left(\frac{\sqrt{kd}}{\sigma/3}\right) + \sigma^2}}.$$

or

$$\lambda_{\min}(\nabla^2 g(v)) < -\frac{1}{24\left(4d\log\left(\frac{\sqrt{kd}}{\sigma/3}\right) + \sigma^2\right)}.$$

This means that, convergence to a $\rho$-second-order stationary point for $g$ implies:

$$\frac{\sigma}{12\sqrt{4d\log\left(\frac{\sqrt{kd}}{\sigma/3}\right) + \sigma^2}} < \rho,$$

$$\frac{1}{24\left(4d\log\left(\frac{\sqrt{kd}}{\sigma/3}\right) + \sigma^2\right)} < \sqrt{K\rho}.$$

We will use the second inequality. First, we observe that:

$$\frac{1}{24\left(12\frac{\sqrt{kd}}{\sigma} + \sigma^2\right)} \leq \frac{1}{24\left(4d\log\left(\frac{\sqrt{kd}}{\sigma/3}\right) + \sigma^2\right)}$$

Therefore, we have:

$$\frac{1}{\sqrt{K\rho}} < 288\frac{\sqrt{kd}}{\sigma} + 24\sigma^2$$

Because of our analysis we have that at all times during the optimization $\sigma^2 < 1$ (refer to the regularized objective function in Equation 4. In addition to that, we assume that $d \geq 2, k \geq 1$, which are both natural assumptions. This in turn implies that:

$$\frac{1}{\sqrt{K\rho}} < 576\frac{\sqrt{kd}}{\sigma}$$

$$\frac{1}{576d\sqrt{K\rho}} < \frac{3\sqrt{kd}}{\sigma}$$

$$\frac{1}{576d\sqrt{K\rho kd}} < \frac{1}{\sigma}$$

$$\sigma^2 < 576^2 K\rho kd^3.$$

Finally, substituting in for the Lipschitz constant $K$, we get:

$\sigma^2 < \text{poly}(n, k, d) \cdot \rho^{O(1)}$.

This implies that $\sigma^2$ can become arbitrarily small for an appropriate choice of $\rho$.

$\square$

## A.4 Proof of Lemma 4

**Lemma.** *Given $n$ unit vectors in $\mathbb{R}^d$ and a target dimension $k$, choose $\varepsilon = O(\sqrt{\log n/k})$ such that distribution $A \sim N(\boldsymbol{M}, \sigma^2)$ satisfies the JL guarantee with distortion $\varepsilon$ with probability $1/6$. Then using matrix $\boldsymbol{M}$ instead of sampling from $A$ retains the JL guarantee with a threshold increased by at most $\mathrm{poly}(\sigma, 1/k)$.*

*Proof.* We start with the assumption that $\frac{1}{k}\|Ax\|_2^2 \in (1 - \varepsilon, 1 + \varepsilon)$ with probability at least $\frac{1}{6}$.

Expressing $A$ as $A = \boldsymbol{M} + Z$ where $Z \sim N(\boldsymbol{0}, \sigma^2)$, and from the JL lemma, we can select $\varepsilon_0$ such that

$$\frac{1}{k}\|Zx\|_2^2 \in [\sigma^2(1 - \varepsilon_0), \sigma^2(1 + \varepsilon_0)],$$

with probability at least $\frac{6}{7}$. This ensures there exists an overlap where both inequalities for $A$ and $Z$ hold simultaneously.

Our goal is to determine how much worse the distortion becomes when using $\boldsymbol{M}$ instead of sampling from the distribution $A$.

Using the triangle inequality we have:

$$\frac{1}{k}\|\boldsymbol{M}x\|_2 = \frac{1}{k}\|\boldsymbol{M}x + Zx - Zx\|_2 \leq \frac{1}{k}\|\boldsymbol{M}x + Zx\|_2 + \frac{1}{k}\|Zx\|_2 = \frac{1}{k}\|Ax\|_2 + \frac{1}{k}\|Zx\|_2,$$

which by squaring both sides and using the JL guarantee for $A$ and $Z$, we obtain:

$$\frac{1}{k}\|\boldsymbol{M}x\|_2^2 \leq \frac{1}{k}\|Ax\|_2^2 + \frac{2}{k^2}\|Ax\|_2\|Zx\|_2 + \frac{1}{k}\|Zx\|_2^2$$
$$\leq 1 + \varepsilon + \frac{2\sigma}{k}\sqrt{1 + \varepsilon}\sqrt{1 + \varepsilon_0} + \sigma^2(1 + \varepsilon_0)$$
$$\leq 1 + \varepsilon + \frac{2\sqrt{2}\sigma}{k}\sqrt{1 + \varepsilon} + 2\sigma^2.$$

For the lower bound, using the Cauchy-Schwarz inequality and the JL guarantee for $A$ and $Z$, we have:

$$\frac{1}{k}\|\boldsymbol{M}x\|_2^2 \geq \frac{1}{2k}\|\boldsymbol{M}x + Zx\|_2^2 - \frac{1}{k}\|Zx\|_2^2$$
$$= \frac{1}{2k}\|Ax\|_2^2 - \frac{1}{k}\|Zx\|_2^2$$
$$\geq 1/2(1 - \varepsilon) - \sigma^2(1 + \varepsilon_0)$$
$$\geq 1/2(1 - \varepsilon) - \sigma^2,$$

Combining these results, we observe that replacing $A$ with $\boldsymbol{M}$ maintains the JL guarantee with an increased distortion threshold, bounded by at most $\mathrm{poly}(\sigma, 1/k)$.

$\square$

## A.5 Proof of Lemma 5

*Proof.* We use Taylor's theorem on the smoothness of $g$ and get:

**Lemma.** *If $\|\nabla g(x_t)\|_2 > \rho$, then for $\nu = \frac{1}{L}$ and $x_{t+1} = x_t - \nu \cdot \nabla g(x_t)$, we have $g(x_{t+1}) \leq g(x_t) - \frac{\nu\rho^2}{2}$.*

$$g(x_{t+1}) \leq g(x_t) + \langle \nabla g(x_t), x_{t+1} - x_t \rangle + \frac{L}{2} \|x_{t+1} - x_t\|_2^2$$

$$= g(x_t) - \nu \|\nabla g(x_t)\|_2^2 + \frac{L\nu^2}{2} \|\nabla g(x_t)\|_2^2$$

$$= g(x_t) - \left(1 - \frac{1}{2} L\nu\right) \nu \rho^2.$$

We can use $\nu = 1/L$ to get the following:

$$g(x_{t+1}) \leq g(x_t) - \frac{1}{2} \nu \rho^2.$$

$\square$

### A.6    Proof of Lemma 6

**Lemma.** *If $\|\nabla g(x_t)\|_2 \leq \rho$ and $\lambda_{min}(\nabla^2 g(x_t)) < -\sqrt{K\rho}$, then for $h = \frac{3\sqrt{\rho}}{K}$ and $x_{t+1} = x_t + hu_1$, where $u_1$ corresponds to the eigenvector of the minimum eigenvalue, we have $g(x_{t+1}) \leq g(x_t) - \frac{3\rho^{1.5}}{4\sqrt{K}}$.*

*Proof.* We use Taylor's theorem on the Lipschitzness of $\nabla^2 g$ [Nesterov, 2013] and get:

$$g(x_{t+1}) \leq g(x_t) + \langle \nabla g(x_t), x_{t+1} - x_t \rangle + \frac{1}{2} \langle \nabla^2 g(x_t)(x_{t+1} - x_t), (x_{t+1} - x_t) \rangle + \frac{K}{6} \|x_{t+1} - x_t\|^3$$

$$= g(x_t) + h\langle \nabla g(x_t), u_1 \rangle + \frac{h^2 \lambda_1}{2} + \frac{h^3 K}{6}$$

$$\leq g(x_t) + h\rho - h^2 \sqrt{K\rho} + \frac{h^3 K}{6}.$$

We can use $h = \frac{3\sqrt{\rho}}{\sqrt{K}}$ to get the following:

$$g(x_{t+1}) \leq g(x_t) - \frac{3\rho\sqrt{\rho}}{2\sqrt{K}}.$$

$\square$

## B    Details of the Experimental Evaluation

We explore the behavior of a distortion optimization process using by minimizing the expected maximum distortion in a given dataset through a series of optimization descent steps. We show that while the Gaussian randomized construction can achieve good enough distortion, our method goes beyond that and achieves almost zero distortion, by taking into account the structure of the data.

### B.1    Proxy for the Objective Function.

To do this, we use the maximum distortion (Equation 1) with a proxy of our objective function (Equation 4) and we aim to minimize:

$$f(\boldsymbol{M}, \sigma^2) = \mathbf{E}_{A \sim N(\boldsymbol{M}, \sigma^2)}[h(A)] + \sigma^2/2. \tag{8}$$

To do this we use the gradient of Equation 8 with respect to the parameters of interest $\theta = (\boldsymbol{M}, \sigma^2)$:

$$\nabla_{(\boldsymbol{M}, \sigma^2)} \mathbf{E}_{A \sim N(\boldsymbol{M}, \sigma^2)}[h(A)] + \sigma^2/2. \tag{9}$$

We can approximate the expectation by taking $y_1, \ldots, y_N$ samples drawn from $N(\boldsymbol{M}, \sigma^2)$ and using Monte Carlo sampling we get the approximate gradient:

$$\nabla_{(\boldsymbol{M}, \sigma^2)} f(\boldsymbol{M}, \sigma^2) \approx \frac{1}{N} \sum_{i=1}^{N} h(y_i).$$

### B.2 Methodology of the Simulation.

We generate a unit norm synthetic dataset of $n = 100$ data points in $d = 500$ dimensions and our goal is to project these into $k = 30$ dimensions while minimizing $f$. We run our optimization for 5000 iterations, using the Adam optimizer [Kingma and Ba, 2014] with a learning rate of 0.01 and batch size $N = 20$. At every iteration, we calculate the maximum distortion and store it. We demonstrate that, through this procedure, the model consistently reduces the distortion and the variance, thus converging to a deterministic solution sampler.

We compare our method with the Gaussian random construction, that is we draw $Z \sim N(\boldsymbol{0}, 1)$ and calculate $\|Zx\|_2^2$. To have a fair evaluation we draw 1000 such matrices and calculate the mean and the minimum distortion obtained.

Our results show that we consistently learn a matrix with close to optimal distortion, that is $\|\boldsymbol{M}x\|_2^2 \approx 1$, while the randomized construction achieves an average value of $\|Z_{\text{avg}}x\|_2^2 \approx 2$ and minimum value $\|Z_{\min}x\|^2 \approx 1.6$.

## C Proving Smoothness and Hessian Lipschitzness

Denote by $\mu_1, \ldots, \mu_k$, the rows of matrix $\boldsymbol{M}$ and $A_1, \ldots, A_k$ the rows of the Gaussian random matrix $A$. Then we can write the objective function:

$$g(\boldsymbol{M}, \sigma^2) = \sum_{j=1}^{n} \Pr\left(\langle A_1, x_j\rangle^2 + \cdots + \langle A_k, x_j\rangle^2 \notin [k(1-\varepsilon), k(1+\varepsilon)]\right) + \frac{\sigma^2}{2}$$

$$= \sum_{j=1}^{n} \Pr\left[\chi_1^2\left(\delta_{1,j} = \frac{\langle \mu_1, x_j\rangle^2}{\sigma^2}\right) + \cdots + \chi_1^2\left(\delta_{k,j} = \frac{\langle \mu_k, x_j\rangle^2}{\sigma^2}\right) \notin R_1\right] + \frac{\sigma^2}{2}$$

$$= \sum_{j=1}^{n} \Pr\left[\chi_k^2(\delta_j) \notin [k(1-\varepsilon)/\sigma^2, k(1+\varepsilon)]\right] + \frac{\sigma^2}{2}$$

$$= \sum_{j=1}^{n}\left[\int_0^{\frac{k(1-\varepsilon)}{\sigma^2}} f_{k,\delta_j}(z)dz + \int_{\frac{k(1+\varepsilon)}{\sigma^2}}^{\infty} f_{k,\delta_j}(z)dz\right] + \frac{\sigma^2}{2}$$

$$= \sum_{j=1}^{n}\left[1 + F_{k,\delta_j}\left(\frac{k(1-\varepsilon)}{\sigma^2}\right) - F_{k,\delta_j}\left(\frac{k(1+\varepsilon)}{\sigma^2}\right)\right] + \frac{\sigma^2}{2}, \tag{10}$$

where $f_{k,\delta_j}, F_{k,\delta_j}$ are the pdf and cdf of $\chi_k^2$, the non-central chi-squared distribution with $k$ degrees of freedom and $\delta_j = \dfrac{\langle \mu_1, x_j\rangle^2 + \cdots + \langle \mu_k, x_j\rangle^2}{\sigma^2}$ as the non centrality parameter, respectively.

Note that, instead of considering the $k \times d$ mean variables directly, it is simpler to reframe the problem. Specifically, we can view the function $g$ in terms of the inner product variables $v_1, \ldots, v_k$ and $\sigma^2$, where $v_i = \langle \mu_i, x\rangle$ and $\mu_i$ is the $i$-th row of the matrix $\boldsymbol{M}$.

Therefore, our problem reduces to proving the Lipschitz continuity of the Gradient and Hessian continuity with respect to these new variables. This approach simplifies the calculations significantly.

To establish that the Gradient and Hessian is Lipschitz continuous, we examine the case for $j = 1$. The extension to $j = n$ can be handled through summation. Let $\tau = \sigma^2$. Then, we have $\delta = \dfrac{\|v\|^2}{\tau} = \dfrac{v_1^2 + \cdots + v_k^2}{\tau}$. Our function from Equation 10 becomes

$$g(v_1, \ldots, v_k, \tau) = 1 + F_{k,\delta}\left(\frac{k(1-\varepsilon)}{\tau}\right) - F_{k,\delta}\left(\frac{k(1+\varepsilon)}{\tau}\right) + \tau/2,$$

where $F_{k,\delta}(x) = e^{-\delta/2} \sum\limits_{j=0}^{\infty} \frac{(\delta/2)^j}{j!} Q(x; k+2j)$, with $Q(x; k) = \frac{\gamma(k/2, x/2)}{\Gamma(k/2)}$ and $\gamma(y, t)$ is the lower incomplete gamma function.

Notice that taking the derivative gives an extra part with increased degrees of freedom $\frac{\partial F_{\delta,k}(x)}{\partial \delta} = -1/2 F_{\delta,k}(x) + 1/2 F_{\delta,k+2}(x)$.

## C.1 Gradient Lipschitzness.

Denote by $D$ the elements of $\nabla g$, and note that in this section to simplify notation we use $\|\cdot\|$ to represent $\|\cdot\|_2$. Then the derivatives are:

$$D_{v_i}(v_1, \ldots, v_k, \tau) = -\frac{v_i}{\tau} F_{k,\delta}\left(\frac{k(1-\varepsilon)}{\tau}\right) + \frac{v_i}{\tau} F_{k+2,\delta}\left(\frac{k(1-\varepsilon)}{\tau}\right)$$
$$+ \frac{v_i}{\tau} F_{k,\delta}\left(\frac{k(1+\varepsilon)}{\tau}\right) - \frac{v_i}{\tau} F_{k+2,\delta}\left(\frac{k(1+\varepsilon)}{\tau}\right).$$

$$D_{\tau}(v_1, \ldots, v_k, \tau) = \frac{\|v\|^2}{2\tau^2} F_{k,\delta}\left(\frac{k(1-\varepsilon)}{\tau}\right) - \frac{\|v\|^2}{2\tau^2} F_{k+2,\delta}\left(\frac{k(1-\varepsilon)}{\tau}\right) - \frac{k(1-\varepsilon)}{\tau^2} f_{k,\delta}\left(\frac{k(1-\varepsilon)}{\tau}\right)$$
$$- \frac{\|v\|^2}{2\tau^2} F_{k,\delta}\left(\frac{k(1+\varepsilon)}{\tau}\right) + \frac{\|v\|^2}{2\tau^2} F_{k+2,\delta}\left(\frac{k(1+\varepsilon)}{\tau}\right) + \frac{k(1+\varepsilon)}{\tau^2} f_{k,\delta}\left(\frac{k(1+\varepsilon)}{\tau}\right).$$

To prove Gradient Lipschitzness of $g$ we can bound the Frobenius norm of the Hessian $\nabla^2 g$. Denote by $D^2$ the elements of $\nabla^2 g$. Below we calculate and bound all the derivatives:

$$D^2_{v_i,v_i} g(v_1, \ldots, v_k, \tau) = \left[-\frac{1}{\tau} + \frac{v_i^2}{\tau^2}\right] F_{k,\delta}\left(\frac{k(1-\varepsilon)}{\tau}\right) + \left[-\frac{2v_i^2}{\tau^2} + \frac{1}{\tau}\right] F_{k+2,\delta}\left(\frac{k(1-\varepsilon)}{\tau}\right)$$
$$+ \frac{v_i^2}{\tau^2} F_{k+4,\delta}\left(\frac{k(1-\varepsilon)}{\tau}\right) + \left[\frac{1}{\tau} - \frac{v_i^2}{\tau^2}\right] F_{k,\delta}\left(\frac{k(1+\varepsilon)}{\tau}\right)$$
$$+ \left[\frac{2v_i^2}{\tau^2} - \frac{1}{\tau}\right] F_{k+2,\delta}\left(\frac{k(1+\varepsilon)}{\tau}\right) - \frac{v_i^2}{\tau^2} F_{k+4,\delta}\left(\frac{k(1+\varepsilon)}{\tau}\right).$$

Here we used the triangle inequality and the fact that the cdf is bounded by 1. Notice how there is no dependency on $\|v\|$ since the range we are integrating over with the cumulative distribution functions is independent of $\|v\|$. Consequently, if $\|v\|$ is large, the probability becomes exponentially small. Thus, we can bound this by:

$$|D^2_{v_i,v_i} g(v_1, \ldots, v_k, \tau)| \le \frac{4}{\tau} + \frac{8}{\tau^2} \le O\left(\frac{1}{\tau^2}\right).$$

Next, we have:

$$D^2_{v_i,v_j} g(v_1, \ldots, v_k, \tau) = \frac{v_i v_j}{\tau^2} F_{k,\delta}\left(\frac{k(1-\varepsilon)}{\tau}\right) - \frac{2v_i v_j}{\tau^2} F_{k+2,\delta}\left(\frac{k(1-\varepsilon)}{\tau}\right) + \frac{v_i v_j}{\tau^2} F_{k+4,\delta}\left(\frac{k(1-\varepsilon)}{\tau}\right)$$
$$- \frac{v_i v_j}{\tau^2} F_{k,\delta}\left(\frac{k(1+\varepsilon)}{\tau}\right) + \frac{2v_i v_j}{\tau^2} F_{k+2,\delta}\left(\frac{k(1+\varepsilon)}{\tau}\right) - \frac{v_i v_j}{\tau^2} F_{k+4,\delta}\left(\frac{k(1+\varepsilon)}{\tau}\right).$$

Thus, we can bound this by:

$$|D^2_{v_i,v_j}g(v_1,\ldots,v_k,\tau)| \leq \frac{8v_iv_j}{\tau^2} \leq O\left(\frac{1}{\tau^2}\right).$$

Next, we have:

$$
\begin{aligned}
D^2_{v_i,\tau}g(v_1,\ldots,v_k,\tau) =& \left(\frac{v_i}{\tau^2} - \frac{v_i\|v\|^2}{\tau^3}\right)F_{k,\delta}\left(\frac{k(1-\varepsilon)}{\tau}\right) + \left(\frac{2v_i\|v\|^2}{\tau^3} - \frac{v_i}{\tau^2}\right)F_{k+2,\delta}\left(\frac{k(1-\varepsilon)}{\tau}\right) \\
&- \frac{v_i\|v\|^2}{\tau^3}F_{k+4,\delta}\left(\frac{k(1-\varepsilon)}{\tau}\right) + \frac{v_ik(1-\varepsilon)}{\tau^3}f_{k,\delta}\left(\frac{k(1-\varepsilon)}{\tau}\right) \\
&- \frac{v_ik(1-\varepsilon)}{\tau^3}f_{k+2,\delta}\left(\frac{k(1-\varepsilon)}{\tau}\right)\left(-\frac{v_i}{\tau^2} + \frac{v_i\|v\|^2}{\tau^3}\right)F_{k,\delta}\left(\frac{k(1+\varepsilon)}{\tau}\right) \\
&+ \left(-\frac{2v_i\|v\|^2}{\tau^3} + \frac{2v_i}{\tau^2}\right)F_{k+2,\delta}\left(\frac{k(1+\varepsilon)}{\tau}\right) + \frac{v_i\|v\|^2}{\tau^3}F_{k+4,\delta}\left(\frac{k(1+\varepsilon)}{\tau}\right) \\
&- \frac{v_ik(1+\varepsilon)}{\tau^3}f_{k,\delta}\left(\frac{k(1+\varepsilon)}{\tau}\right) + \frac{v_ik(1+\varepsilon)}{\tau^2}f_{k+2,\delta}\left(\frac{k(1+\varepsilon)}{\tau}\right).
\end{aligned}
$$

Thus, we can bound this by:

$$|D^2_{v_i,\tau}g(v_1,\ldots,v_k,\tau)| \leq \frac{4v_i}{\tau^2} + \frac{8v_i\|v\|^2}{\tau^3} + \frac{2v_ik(1-\varepsilon)}{\tau^3} + \frac{2v_ik(1+\varepsilon)}{\tau^3} \leq O\left(\frac{k}{\tau^3}\right).$$

Next, we have:

$$
\begin{aligned}
D^2_{\tau,\tau}g(v_1,\ldots,v_k,\tau) =& \left(-\frac{\|v\|^2}{\tau^3} + \frac{\|v\|^4}{4\tau^4}\right)F_{k,\delta}\left(\frac{k(1-\varepsilon)}{\tau}\right) + \left(-\frac{\|v\|^4}{2\tau^4} + \frac{\|v\|^2}{\tau^3}\right)F_{k+2,\delta}\left(\frac{k(1-\varepsilon)}{\tau}\right) \\
&+ \frac{\|v\|^4}{4\tau^4}F_{k+4,\delta}\left(\frac{k(1-\varepsilon)}{\tau}\right) \\
&+ \left(-\frac{\|v\|^2k(1-\varepsilon)}{2\tau^4} + \frac{2k(1-\varepsilon)}{\tau^3} - \frac{(k(1-\varepsilon))^2}{2\tau^4}\right)f_{k,\delta}\left(\frac{k(1-\varepsilon)}{\tau}\right) \\
&+ \frac{\|v\|^2k(1-\varepsilon)}{2\tau^4}f_{k+2,\delta}\left(\frac{k(1-\varepsilon)}{\tau}\right) \\
&+ \frac{k(1-\varepsilon)}{\tau^3}e^{-\delta/2}\sum_{j=0}^{\infty}\frac{(\delta/2)^j}{j!}((l+2j)/2-1)f_{k+2j}\left(\frac{k(1-\varepsilon)}{\tau}\right) \\
&+ \left(\frac{\|v\|^2}{\tau^3} - \frac{\|v\|^4}{4\tau^4}\right)F_{k,\delta}\left(\frac{k(1+\varepsilon)}{\tau}\right) + \left(\frac{\|v\|^4}{2\tau^4} - \frac{\|v\|^2}{\tau^3}\right)F_{k+2,\delta}\left(\frac{k(1+\varepsilon)}{\tau}\right) \\
&- \frac{\|v\|^4}{4\tau^4}F_{k+4,\delta}\left(\frac{k(1+\varepsilon)}{\tau}\right) \\
&+ \left(\frac{\|v\|^2k(1+\varepsilon)}{2\tau^4} - \frac{2k(1+\varepsilon)}{\tau^3} + \frac{(k(1+\varepsilon))^2}{2\tau^4}\right)f_{k,\delta}\left(\frac{k(1+\varepsilon)}{\tau}\right) \\
&- \frac{\|v\|^2k(1+\varepsilon)}{2\tau^4}f_{k+2,\delta}\left(\frac{k(1+\varepsilon)}{\tau}\right) \\
&- \frac{k(1+\varepsilon)}{\tau^3}e^{-\delta/2}\sum_{j=0}^{\infty}\frac{(\delta/2)^j}{j!}((l+2j)/2-1)f_{k+2j}\left(\frac{k(1+\varepsilon)}{\tau}\right) + \frac{1}{2}.
\end{aligned}
$$

Thus, we can bound this by:

$$|D^2_{\tau,\tau}g(v_1,\ldots,v_k,\tau)| \le \frac{4\|v\|^2}{\tau^3} + \frac{2\|v\|^4}{\tau^4} + \frac{\|v\|^2 k(1-\varepsilon)}{\tau^4} + \frac{3k(1-\varepsilon)}{\tau^3} + \frac{(k(1-\varepsilon))^2}{2\tau^4}$$
$$+ \frac{\|v\|^2 k(1+\varepsilon)}{\tau^4} + \frac{3k(1+\varepsilon)}{\tau^3} + \frac{(k(1+\varepsilon))^2}{2\tau^4}$$
$$\le O\left(\frac{k\varepsilon^2}{\tau^4}\right).$$

Overall, for $D^2_{v_i,v_i}g$, we have:

$$\xi_{v_i,v_i} = \max_{v_1,\ldots,v_k,\tau} \|D^2_{v_i,v_i}g(v_1,\ldots,v_k,\tau)\|$$
$$\le \min_\tau(1/\tau^2).$$

Similarly, for $D^2_{v_i,v_j}g$, we obtain $\xi_{v_i,v_j} \le \min_\tau(1/\tau^2)$ for $D^2_{v_i,\tau}g$, we obtain $\xi_{v_i,\tau} \le \min_\tau(k/\tau^3)$ and for $D^2_{\tau,\tau}$, we obtain $\xi_{\tau,\tau} \le \min_\tau(k^2\varepsilon^2/\tau^4)$.

Substituting $M$ and $\sigma^2$ back, and assuming that the minimum value for $\sigma^2$ we allow is $\sigma_0^2$ we have that the Frobenius norm of the Hessian is bounded from:

$$\|\nabla^2 g\|_F^2 \le \sum_{i=1}^k \xi_{v_i,v_i}^2 + \xi_{\sigma^2,\sigma^2}^2 + 2\sum_{i\ne j} \xi_{v_i,v_j}^2 + 2\sum_{i=1}^k \xi_{v_i,\sigma^2}^2$$
$$\le k\xi_{v_i,v_i}^2 + \xi_{\sigma^2,\sigma^2}^2 + 2(k \times d - 2k - 1)\xi_{v_i,v_j}^2 + 2k\xi_{v_i,\sigma^2}^2.$$

Therefore we get:

$$\|\nabla^2 g\|_F \le O\left(\frac{k}{\sigma_0^4}\right) + O\left(\frac{k^2\varepsilon^2}{\sigma_0^8}\right) + O\left(\frac{dk}{\sigma_0^4}\right) + O\left(\frac{k}{\sigma_0^6}\right) \le \text{poly}\left(k,\varepsilon,d,1/\sigma_0\right).$$

Finally, since we are summing over $n$ data points the smoothness Lipschitz constant is $\text{poly}(n,k,\varepsilon,d,1/\sigma_0)$

### C.2 Hessian Lipschitzness.

To prove Hessian Lipschitzness of $g$ we will use the definition and the third-order derivatives. Denote by $D^3$ the elements of $\nabla^3 g$. Below we calculate and bound all the third-order derivatives:

$$D^3_{v_i,v_i,v_i}g(v_1,\ldots,v_k,\tau) = \left[\frac{3v_i}{\tau^2} - \frac{v_i^3}{\tau^3}\right] F_{k,\delta}\left(\frac{k(1-\varepsilon)}{\tau}\right) + \left[-\frac{6v_i}{\tau^2} + \frac{3v_i^3}{\tau^3}\right] F_{k+2,\delta}\left(\frac{k(1-\varepsilon)}{\tau}\right)$$
$$+ \left[\frac{3v_i}{\tau^2} - \frac{3v_i^3}{\tau^3}\right] F_{k+4,\delta}\left(\frac{k(1-\varepsilon)}{\tau}\right) + \frac{v_i^3}{\tau^3} F_{k+6,\delta}\left(\frac{k(1-\varepsilon)}{\tau}\right)$$
$$+ \left[-\frac{3v_i}{\tau^2} + \frac{v_i^3}{\tau^3}\right] F_{k,\delta}\left(\frac{k(1+\varepsilon)}{\tau}\right) + \left[\frac{6v_i}{\tau^2} - \frac{3v_i^3}{\tau^3}\right] F_{k+2,\delta}\left(\frac{k(1+\varepsilon)}{\tau}\right)$$
$$+ \left[-\frac{3v_i}{\tau^2} + \frac{3v_i^3}{\tau^3}\right] F_{k+4,\delta}\left(\frac{k(1+\varepsilon)}{\tau}\right) - \frac{v_i^3}{\tau^3} F_{k+6,\delta}\left(\frac{k(1+\varepsilon)}{\tau}\right).$$

Thus, we can bound this:

$$|D^3_{v_i,v_i,v_i}g(v_1,\ldots,v_k,\tau)| \le \frac{24v_i}{\tau^2} + \frac{16v_i^3}{\tau^3} \le \left(\frac{1}{\tau^3}\right).$$

Here we used the triangle inequality and the fact that the cdf is bounded by 1. Notice how there is no dependency on $\|v\|$ since the range we are integrating over with the cumulative distribution functions is independent of $\|v\|$. Consequently, if $\|v\|$ is large, the probability becomes exponentially small.

Next, we have:

$$
\begin{aligned}
D^3_{v_i,v_j,v_j} g(v_1,\ldots,v_k,\tau) &= \left[\frac{v_i}{\tau^2} - \frac{v_i v_j^2}{\tau^3}\right] F_{k,\delta}\left(\frac{k(1-\varepsilon)}{\tau}\right) + \left[-\frac{2v_i}{\tau^2} + \frac{3v_i v_j^2}{\tau^3}\right] F_{k+2,\delta}\left(\frac{k(1-\varepsilon)}{\tau}\right) \\
&+ \left[\frac{v_i}{\tau^2} - \frac{3v_i v_j^2}{\tau^3}\right] F_{k+4,\delta}\left(\frac{k(1-\varepsilon)}{\tau}\right) + \frac{v_i v_j^2}{\tau^3} F_{k+6,\delta}\left(\frac{k(1-\varepsilon)}{\tau}\right) \\
&+ \left[-\frac{v_i}{\tau^2} + \frac{v_i v_j^2}{\tau^3}\right] F_{k,\delta}\left(\frac{k(1+\varepsilon)}{\tau}\right) + \left[\frac{2v_i}{\tau^2} - \frac{3v_i v_j^2}{\tau^3}\right] F_{k+2,\delta}\left(\frac{k(1+\varepsilon)}{\tau}\right) \\
&+ \left[-\frac{v_i}{\tau^2} + \frac{3v_i v_j^2}{\tau^3}\right] F_{k+4,\delta}\left(\frac{k(1+\varepsilon)}{\tau}\right) - \frac{v_i v_j^2}{\tau^3} F_{k+6,\delta}\left(\frac{k(1+\varepsilon)}{\tau}\right).
\end{aligned}
$$

Thus, we can bound this by:

$$
|D^3_{v_i,v_j,v_j} g(v_1,\ldots,v_k,\sigma)| \le \frac{8v_i}{\tau^2} + \frac{16 v_i v_j^2}{\tau^3} \le \left(\frac{1}{\tau^3}\right).
$$

Next, we have:

$$
\begin{aligned}
D^3_{v_i,v_j,v_k} g(v_1,\ldots,v_k,\tau) &= -\frac{v_i v_j v_k}{\tau^3} F_{k,\delta}\left(\frac{k(1-\varepsilon)}{\tau}\right) + \frac{3v_i v_j v_k}{\tau^3} F_{k+2,\delta}\left(\frac{k(1-\varepsilon)}{\tau}\right) \\
&- \frac{3v_i v_j v_k}{\tau^3} F_{k+4,\delta}\left(\frac{k(1-\varepsilon)}{\tau}\right) + \frac{v_i v_j v_k}{\tau^3} F_{k+6,\delta}\left(\frac{k(1-\varepsilon)}{\tau}\right) \\
&+ \frac{v_i v_j v_k}{\tau^3} F_{k,\delta}\left(\frac{k(1+\varepsilon)}{\tau}\right) - \frac{3v_i v_j v_k}{\tau^3} F_{k+2,\delta}\left(\frac{k(1+\varepsilon)}{\tau}\right) \\
&+ \frac{3v_i v_j v_k}{\tau^3} F_{k+4,\delta}\left(\frac{k(1+\varepsilon)}{\tau}\right) - \frac{v_i v_j v_k}{\tau^3} F_{k+6,\delta}\left(\frac{k(1+\varepsilon)}{\tau}\right).
\end{aligned}
$$

Thus, we can bound this by:

$$
|D^3_{v_i,v_j,v_k} g(v_1,\ldots,v_k,\tau)| \le \frac{16 v_i v_j v_k}{\tau^3} \le \left(\frac{1}{\tau^3}\right).
$$

Next, we have:

$$D^3_{v_i,\tau,v_i}g(v_1,\ldots,v_k,\tau) = \left[\frac{1}{\tau^2} - \frac{\|v\|^2}{\tau^3} - \frac{3v_i^2}{\tau^3} + \frac{v_i^2\|v\|^2}{\tau^4}\right] F_{k,\delta}\left(\frac{k(1-\varepsilon)}{\tau}\right)$$

$$+ \left[\frac{6v_i^2}{\tau^3} - \frac{3v_i^2\|v\|^2}{\tau^4} + \frac{2\|v\|^2}{\tau^3} - \frac{1}{\tau^2}\right] F_{k+2,\delta}\left(\frac{k(1-\varepsilon)}{\tau}\right)$$

$$+ \left[\frac{3v_i^2\|v\|^2}{\tau^4} - \frac{3v_i^2}{\tau^3} - \frac{\|v\|^2}{\tau^3}\right] F_{k+4,\delta}\left(\frac{k(1-\varepsilon)}{\tau}\right) - \frac{v_i^2\|v\|^2}{\tau^4}F_{k+6,\delta}\left(\frac{k(1-\varepsilon)}{\tau}\right)$$

$$+ \left(\frac{k(1-\varepsilon)}{\tau^3} - \frac{v_i^2 k(1-\varepsilon)}{\tau^4}\right) f_{k,\delta}\left(\frac{k(1-\varepsilon)}{\tau}\right)$$

$$+ \left(-\frac{k(1-\varepsilon)}{\tau^3} + \frac{2v_i^2 k(1-\varepsilon)}{\tau^4}\right) f_{k+2,\delta}\left(\frac{k(1-\varepsilon)}{\tau}\right)$$

$$- \left(\frac{v_i^2 k(1-\varepsilon)}{\tau^4}\right) f_{k+4,\delta}\left(\frac{k(1-\varepsilon)}{\tau}\right)$$

$$+ \left[-\frac{1}{\tau^2} + \frac{\|v\|^2}{\tau^3} + \frac{3v_i^2}{\tau^3} - \frac{v_i^2\|v\|^2}{\tau^4}\right] F_{k,\delta}\left(\frac{k(1+\varepsilon)}{\tau}\right)$$

$$+ \left[-\frac{6v_i^2}{\tau^3} + \frac{3v_i^2\|v\|^2}{\tau^4} - \frac{2\|v\|^2}{\tau^3} + \frac{1}{\tau^2}\right] F_{k+2,\delta}\left(\frac{k(1+\varepsilon)}{\tau}\right)$$

$$+ \left[-\frac{3v_i^2\|v\|^2}{\tau^4} + \frac{3v_i^2}{\tau^3} + \frac{\|v\|^2}{\tau^3}\right] F_{k+4,\delta}\left(\frac{k(1+\varepsilon)}{\tau}\right) + \frac{v_i^2\|v\|^2}{\tau^4}F_{k+6,\delta}\left(\frac{k(1+\varepsilon)}{\tau}\right)$$

$$+ \left(-\frac{k(1+\varepsilon)}{\tau^3} + \frac{v_i^2 k(1+\varepsilon)}{\tau^4}\right) f_{k,\delta}\left(\frac{k(1+\varepsilon)}{\tau}\right)$$

$$+ \left(\frac{k(1+\varepsilon)}{\tau^3} - \frac{2v_i^2 k(1+\varepsilon)}{\tau^4}\right) f_{k+2,\delta}\left(\frac{k(1+\varepsilon)}{\tau}\right)$$

$$+ \left(\frac{v_i^2 k(1+\varepsilon)}{\tau^4}\right) f_{k+4,\delta}\left(\frac{k(1+\varepsilon)}{\tau}\right).$$

Thus, we can bound this by:

$$|D^3_{v_i,\tau,v_i}g(v_1,\ldots,v_k,\tau)| \le \frac{4}{\tau^2} + \frac{8\|v\|^2}{\tau^3} + \frac{24v_i^2}{\tau^3} + \frac{16v_i^2\|v\|^2}{\tau^4} +$$

$$+ \frac{2k(1-\varepsilon)}{\tau^3} + \frac{4v_i^2 k(1-\varepsilon)}{\tau^4}$$

$$+ \frac{2k(1+\varepsilon)}{\tau^3} + \frac{4v_i^2 k(1+\varepsilon)}{\tau^4} \le \left(\frac{k}{\tau^4}\right).$$

Next, we have:

$$
\begin{aligned}
D^3_{v_i,\tau,v_j}g(v_1,\ldots,v_k,\tau) =& \left[-\frac{3v_iv_j}{\tau^3} + \frac{v_iv_j\|v\|^2}{\tau^4}\right] F_{k,\delta}\left(\frac{k(1-\varepsilon)}{\tau}\right) \\
&+ \left[\frac{6v_iv_j}{\tau^3} - \frac{3v_iv_j\|v\|^2}{\tau^4}\right] F_{k+2,\delta}\left(\frac{k(1-\varepsilon)}{\tau}\right) \\
&+ \left[\frac{3v_iv_j\|v\|^2}{\tau^4} - \frac{3v_iv_j}{\tau^3}\right] F_{k+4,\delta}\left(\frac{k(1-\varepsilon)}{\tau}\right) \\
&- \frac{v_iv_j\|v\|^2}{\tau^4} F_{k+6,\delta}\left(\frac{k(1-\varepsilon)}{\tau}\right) \\
&+ \frac{2v_iv_jk(1-\varepsilon)}{\tau^4} f_{k,\delta}\left(\frac{k(1-\varepsilon)}{\tau}\right) - \frac{4v_iv_jk(1-\varepsilon)}{\tau^4} f_{k+2,\delta}\left(\frac{k(1-\varepsilon)}{\tau}\right) \\
&+ \frac{2v_iv_jk(1-\varepsilon)}{\tau^4} f_{k+4,\delta}\left(\frac{k(1-\varepsilon)}{\tau}\right) \\
&+ \left[\frac{3v_iv_j}{\tau^3} - \frac{v_iv_j\|v\|^2}{\tau^4}\right] F_{k,\delta}\left(\frac{k(1+\varepsilon)}{\tau}\right) \\
&+ \left[-\frac{6v_iv_j}{\tau^3} + \frac{3v_iv_j\|v\|^2}{\tau^4}\right] F_{k+2,\delta}\left(\frac{k(1+\varepsilon)}{\tau}\right) \\
&+ \left[-\frac{3v_iv_j\|v\|^2}{\tau^4} + \frac{3v_iv_j}{\tau^3}\right] F_{k+4,\delta}\left(\frac{k(1+\varepsilon)}{\tau}\right) \\
&+ \frac{v_iv_j\|v\|^2}{\tau^4} F_{k+6,\delta}\left(\frac{k(1+\varepsilon)}{\tau}\right) \\
&- \frac{2v_iv_jk(1+\varepsilon)}{\tau^4} f_{k,\delta}\left(\frac{k(1+\varepsilon)}{\tau}\right) + \frac{4v_iv_jk(1+\varepsilon)}{\tau^4} f_{k+2,\delta}\left(\frac{k(1+\varepsilon)}{\tau}\right) \\
&- \frac{2v_iv_jk(1+\varepsilon)}{\tau^4} f_{k+4,\delta}\left(\frac{k(1+\varepsilon)}{\tau}\right).
\end{aligned}
$$

Thus, we can bound this by:

$$
|D^3_{v_i,\tau,v_j}g(v_1,\ldots,v_k,\tau)| \leq \frac{24v_iv_j}{\tau^3} + \frac{16v_iv_j\|v\|^2}{\tau^4} + \frac{16v_iv_jk(1-\varepsilon)}{\tau^4} + \frac{16v_iv_jk(1+\varepsilon)}{\tau^4} \leq \left(\frac{k}{\tau^4}\right).
$$

Next, we have:

$$D^3_{v_i,\tau,\tau} g(v_1,\ldots,v_k,\tau) = \left[ -\frac{2v_i}{\tau^3} + \frac{2\|v\|^2 v_i}{\tau^4} - \frac{v_i\|v\|}{4\tau^5} \right] F_{k,\delta}\left( \frac{k(1-\varepsilon)}{\tau} \right)$$

$$+ \left[ -\frac{6\|v\|^2 v_i}{\tau^4} + \frac{3v_i\|v\|^4}{4\tau^5} + \frac{2v_i}{\tau^3} \right] F_{k+2,\delta}\left( \frac{k(1-\varepsilon)}{\tau} \right)$$

$$+ \left[ -\frac{v_i\|v\|^4}{2\tau^5} + \frac{2v_i\|v\|^2}{\tau^4} - \frac{v_i\|v\|^4}{4\tau^4} \right] F_{k+4,\delta}\left( \frac{k(1-\varepsilon)}{\tau} \right)$$

$$+ \frac{v_i\|v\|^4}{4\tau^4} F_{k+6,\delta}\left( \frac{k(1-\varepsilon)}{\tau} \right)$$

$$+ \left[ -\frac{3v_i k(1-\varepsilon)}{\tau^4} + \frac{v_i\|v\|^2 k(1-\varepsilon)}{2\tau^5} + \frac{v_i(k(1-\varepsilon))^2}{\tau^5} \right] f_{k,\delta}\left( \frac{k(1-\varepsilon)}{\tau} \right)$$

$$+ \left[ -\frac{v_i\|v\|^2 k(1-\varepsilon)}{\tau^5} + \frac{3v_i k(1-\varepsilon)}{\tau^4} - \frac{v_i(k(1-\varepsilon))^2}{2\tau^5} \right] f_{k+2,\delta}\left( \frac{k(1-\varepsilon)}{\sigma^2} \right)$$

$$+ \frac{v_i\|v\|^2 k(1-\varepsilon)}{2\tau^5} f_{k+4,\delta}\left( \frac{k(1-\varepsilon)}{\tau} \right)$$

$$- \frac{v_i k(1-\varepsilon)}{\tau^4} e^{-\delta/2} \sum_{j=0}^{\infty} \frac{(\delta/2)^j}{j!} ((l+2j)/2 - 1) f_{k+2j}\left( \frac{k(1-\varepsilon)}{\tau} \right)$$

$$+ \frac{v_i k(1-\varepsilon)}{\tau^4} e^{-\delta/2} \sum_{j=0}^{\infty} \frac{(\delta/2)^j}{j!} ((l+2j+2)/2 - 1) f_{k+2j+2}\left( \frac{k(1-\varepsilon)}{\tau} \right)$$

$$+ \left[ \frac{2v_i}{\tau^3} - \frac{2\|v\|^2 v_i}{\tau^4} + \frac{v_i\|v\|}{4\tau^5} \right] F_{k,\delta}\left( \frac{k(1+\varepsilon)}{\tau} \right)$$

$$+ \left[ \frac{6\|v\|^2 v_i}{\tau^4} - \frac{3v_i\|v\|^4}{4\tau^5} - \frac{2v_i}{\tau^3} \right] F_{k+2,\delta}\left( \frac{k(1+\varepsilon)}{\tau} \right)$$

$$+ \left[ \frac{v_i\|v\|^4}{2\tau^5} - \frac{2v_i\|v\|^2}{\tau^4} + \frac{v_i\|v\|^4}{4\tau^4} \right] F_{k+4,\delta}\left( \frac{k(1+\varepsilon)}{\tau} \right)$$

$$- \frac{v_i\|v\|^4}{4\tau^4} F_{k+6,\delta}\left( \frac{k(1+\varepsilon)}{\tau} \right)$$

$$+ \left[ \frac{3v_i k(1+\varepsilon)}{\tau^4} - \frac{v_i\|v\|^2 k(1+\varepsilon)}{2\tau^5} - \frac{v_i(k(1+\varepsilon))^2}{\tau^5} \right] f_{k,\delta}\left( \frac{k(1+\varepsilon)}{\tau} \right)$$

$$+ \left[ \frac{v_i\|v\|^2 k(1+\varepsilon)}{\tau^5} - \frac{3v_i k(1+\varepsilon)}{\tau^4} + \frac{v_i(k(1+\varepsilon))^2}{2\tau^5} \right] f_{k+2,\delta}\left( \frac{k(1+\varepsilon)}{\sigma^2} \right)$$

$$- \frac{v_i\|v\|^2 k(1+\varepsilon)}{2\tau^5} f_{k+4,\delta}\left( \frac{k(1+\varepsilon)}{\tau} \right)$$

$$+ \frac{v_i k(1+\varepsilon)}{\tau^4} e^{-\delta/2} \sum_{j=0}^{\infty} \frac{(\delta/2)^j}{j!} ((l+2j)/2 - 1) f_{k+2j}\left( \frac{k(1+\varepsilon)}{\tau} \right)$$

$$- \frac{v_i k(1+\varepsilon)}{\tau^4} e^{-\delta/2} \sum_{j=0}^{\infty} \frac{(\delta/2)^j}{j!} ((l+2j+2)/2 - 1) f_{k+2j+2}\left( \frac{k(1+\varepsilon)}{\tau} \right).$$

Thus, we can bound this by:

$$|D^3_{v_i,\tau,\tau}g(v_1,\ldots,v_k,\tau)| \leq \frac{8v_i}{\tau^3} + \frac{20\|v\|^2 v_i}{\tau^4} + \frac{v_i\|v\|}{2\tau^5} + \frac{14v_i\|v\|^4}{4\tau^4}$$
$$+ \frac{8v_i k(1-\varepsilon)}{\tau^4} + \frac{2v_i\|v\|^2 k(1-\varepsilon)}{\tau^5} + \frac{3v_i(k(1-\varepsilon))^2}{2\tau^5}$$
$$+ \frac{8v_i k(1+\varepsilon)}{\tau^4} + \frac{2v_i\|v\|^2 k(1+\varepsilon)}{\tau^5} + \frac{3v_i(k(1+\varepsilon))^2}{2\tau^5}$$
$$\leq \left(\frac{k^2\varepsilon^2}{\tau^5}\right).$$

Next, we have:

$$D^3_{\tau,\tau,\tau}g(v_1,\ldots,v_k,\tau) = \left[\frac{3\|v\|^2}{\tau^4} - \frac{3\|v\|^4}{2\tau^5} + \frac{\|v\|^6}{8\tau^6}\right] F_{k,\delta}\left(\frac{k(1-\varepsilon)}{\tau}\right)$$
$$+ \left[\frac{4\|v\|^4}{\tau^5} - \frac{3\|v\|^6}{8\tau^6} - \frac{3\|v\|^2}{\tau^4}\right] F_{k+2,\delta}\left(\frac{k(1-\varepsilon)}{\tau}\right)$$
$$+ \left[\frac{3\|v\|^6}{8\tau^6} - \frac{3\|v\|^4}{2\tau^5}\right] F_{k+4,\delta}\left(\frac{k(1-\varepsilon)}{\tau}\right) - \frac{\|v\|^6}{8\tau^6} F_{k+6,\delta}\left(\frac{k(1-\varepsilon)}{\tau}\right)$$
$$+ \left[\frac{4\|v\|^2 k(1-\varepsilon)}{\tau^5} - \frac{\|v\|^4 k(1-\varepsilon)}{2\tau^6} - \frac{6k(1-\varepsilon)}{\tau^4} + \frac{2(k(1-\varepsilon))^2}{\tau^5} - \frac{\|v\|^2(k(1-\varepsilon))^2}{4\tau^6}\right] f_{k,\delta}\left(\frac{k(1-\varepsilon)}{\tau}\right)$$
$$+ \left[\frac{\|v\|^4 k(1-\varepsilon)}{\tau^6} - \frac{4\|v\|^2 k(1-\varepsilon)}{\tau^5} + \frac{\|v\|^2(k(1-\varepsilon))^2}{4\tau^6}\right] f_{k+2,\delta}\left(\frac{k(1-\varepsilon)}{\tau}\right)$$
$$- \frac{\|v\|^4 k(1-\varepsilon)}{4\tau^6} f_{k+4,\delta}\left(\frac{k(1-\varepsilon)}{\tau}\right)$$
$$+ \left[\frac{\|v\|^2 k(1-\varepsilon)}{2\tau^5} - \frac{3k(1-\varepsilon)}{\tau^4} + \frac{(k(1-\varepsilon))^2}{2\tau^5}\right] e^{-\delta/2} \sum_{j=0}^{\infty} \frac{(\delta/2)^j}{j!}(k+2j-2) f_{k+2j}\left(\frac{k(1-\varepsilon)}{\tau}\right)$$
$$- \frac{\|v\|^2 k(1-\varepsilon)}{2\tau^5} e^{-\delta/2} \sum_{j=0}^{\infty} \frac{(\delta/2)^j}{j!}(l/2+j) f_{k+2+2j}\left(\frac{k(1-\varepsilon)}{\tau}\right)$$
$$- \frac{k(1-\varepsilon)}{\tau^4} e^{-\delta/2} \sum_{j=0}^{\infty} \frac{(\delta/2)^j}{j!}((l+2j)/2-1)^2 f_{k+2j}\left(\frac{k(1-\varepsilon)}{\tau}\right)$$

$$+ \left[ -\frac{3\|v\|^2}{\tau^4} + \frac{3\|v\|^4}{2\tau^5} - \frac{\|v\|^6}{8\tau^6} \right] F_{k,\delta}\left( \frac{k(1+\varepsilon)}{\tau} \right)$$

$$+ \left[ -\frac{4\|v\|^4}{\tau^5} + \frac{3\|v\|^6}{8\tau^6} + \frac{3\|v\|^2}{\tau^4} \right] F_{k+2,\delta}\left( \frac{k(1+\varepsilon)}{\tau} \right)$$

$$+ \left[ -\frac{3\|v\|^6}{8\tau^5} + \frac{3\|v\|^4}{2\tau^5} \right] F_{k+4,\delta}\left( \frac{k(1+\varepsilon)}{\tau} \right) + \frac{\|v\|^6}{8\tau^6} F_{k+6,\delta}\left( \frac{k(1+\varepsilon)}{\tau} \right)$$

$$+ \left[ -\frac{4\|v\|^2 k(1+\varepsilon)}{\tau^5} + \frac{\|v\|^4 k(1+\varepsilon)}{2\tau^6} + \frac{6k(1+\varepsilon)}{\tau^4} - \frac{2(k(1+\varepsilon))^2}{\tau^5} + \frac{\|v\|^2 (k(1+\varepsilon))^2}{4\tau^6} \right] f_{k,\delta}\left( \frac{k(1+\varepsilon)}{\tau} \right)$$

$$+ \left[ -\frac{\|v\|^4 k(1+\varepsilon)}{\tau^6} + \frac{4\|v\|^2 k(1+\varepsilon)}{\tau^5} - \frac{\|v\|^2 (k(1+\varepsilon))^2}{4\tau^6} \right] f_{k+2,\delta}\left( \frac{k(1+\varepsilon)}{\tau} \right)$$

$$+ \frac{\|v\|^4 k(1+\varepsilon)}{4\tau^6} f_{k+4,\delta}\left( \frac{k(1+\varepsilon)}{\tau} \right)$$

$$+ \left[ -\frac{\|v\|^2 k(1-\varepsilon)}{2\tau^5} + \frac{3k(1+\varepsilon)}{\tau^4} - \frac{(k(1+\varepsilon))^2}{2\tau^5} \right] e^{-\delta/2} \sum_{j=0}^{\infty} \frac{(\delta/2)^j}{j!} (k+2j-2) f_{k+2j}\left( \frac{k(1+\varepsilon)}{\tau} \right)$$

$$+ \frac{\|v\|^2 k(1+\varepsilon)}{2\tau^5} e^{-\delta/2} \sum_{j=0}^{\infty} \frac{(\delta/2)^j}{j!} (l/2+j) f_{k+2+2j}\left( \frac{k(1+\varepsilon)}{\tau} \right)$$

$$+ \frac{k(1+\varepsilon)}{\tau^4} e^{-\delta/2} \sum_{j=0}^{\infty} \frac{(\delta/2)^j}{j!} ((l+2j)/2-1)^2 f_{k+2j}\left( \frac{k(1+\varepsilon)}{\tau} \right).$$

Thus, we can bound this by:

$$|D^3_{\tau,\tau,\tau} g(v_1, \ldots, v_k, \tau)| \leq \frac{12\|v\|^2}{\tau^4} + \frac{7\|v\|^4}{\tau^5} + \frac{\|v\|^6}{\tau^5}$$

$$+ \frac{8\|v\|^2 k(1-\varepsilon)}{\tau^5} + \frac{7\|v\|^4 k(1-\varepsilon)}{4\tau^6} + \frac{10k(1-\varepsilon)}{\tau^4} + \frac{5(k(1-\varepsilon))^2}{2\tau^5} + \frac{\|v\|^2 (k(1-\varepsilon))^2}{2\tau^6}$$

$$+ \frac{7\|v\|^4 k(1-\varepsilon)}{4\tau^6}$$

$$+ \frac{8\|v\|^2 k(1+\varepsilon)}{\tau^5} + \frac{7\|v\|^4 k(1+\varepsilon)}{4\tau^6} + \frac{10k(1+\varepsilon)}{\tau^4} + \frac{5(k(1+\varepsilon))^2}{2\tau^5} + \frac{\|v\|^2 (k(1+\varepsilon))^2}{2\tau^6}$$

$$+ \frac{7\|v\|^4 k(1+\varepsilon)}{4\tau^6}$$

$$\leq \left( \frac{k^2 \varepsilon^2}{\tau^6} \right).$$

Overall, for $D^2_{v_i,v_i} g$, we have:

$$\rho_{v_i,v_i} = \max_{v_1,\ldots,v_k,\tau} \|\nabla D^2_{v_i,v_i} g(v_1, \ldots, v_k, \tau)\|$$

$$\leq \max_{v_1,\ldots,v_k,\tau} \{ |D^3_{v_i,v_i,v_i} g(v_1, \ldots, v_k, \tau)| + |D^3_{v_i,v_i,v_j} g(v_1, \ldots, v_k, \tau)| + |D^3_{v_i,v_i,\tau} g(v_1, \ldots, v_k, \tau)| \}$$

$$\leq \min_{\tau} (k/\tau^4).$$

Similarly, for $D^2_{v_i,v_j} g$, we obtain $\rho_{v_i,v_j} \leq \min_{\tau} (k\varepsilon/\tau^4)$ for $D^2_{v_i,\tau} g$, we obtain $\rho_{v_i,\tau} \leq \min_{\tau} (k^2\varepsilon^2/\tau^5)$ and for $D^2_{\tau,\tau}$, we obtain $\rho_{\tau,\tau} \leq \min_{\tau} (k^2\varepsilon^2/\tau^6)$.

Substituting $M$ and $\sigma^2$ back, and assuming that the minimum value for $\sigma^2$ we allow is $\sigma_0^2$ we have:

$$\|H(\boldsymbol{M}_1, \sigma_1^2) - H(\boldsymbol{M}_2, \sigma_2^2)\|^2 = \sum_{i=1}^{k} \left[g_{v_i,v_i}(\boldsymbol{M}_1, \sigma_1^2) - g_{v_i,v_i}(\boldsymbol{M}_2, \sigma_2^2)\right]^2 + \left[g_{\sigma^2,\sigma^2}(\boldsymbol{M}_1, \sigma_1^2) - g_{\sigma^2,\sigma^2}(\boldsymbol{M}_2, \sigma_2^2)\right]^2$$

$$+ \sum_{i \neq j} 2\left[g_{v_i,v_j}(\boldsymbol{M}_1, \sigma_1^2) - g_{v_i,v_j}(\boldsymbol{M}_2, \sigma_2^2)\right]^2 + 2\sum_{i=1}^{k}\left[g_{v_i,\sigma^2}(\boldsymbol{M}_1, \sigma_1^2) - g_{v_i,\sigma^2}(\boldsymbol{M}_2, \sigma_2^2)\right]|^2$$

$$\leq \left(k\rho_{v_1,v_1}^2 + \rho_{\sigma^2,\sigma^2}^2 + 2(k \times d - 2k - 1)\rho_{v_1,v_2}^2 + 2k\rho_{v_1,\sigma^2}^2\right)\left[(\boldsymbol{M}_1 - \boldsymbol{M}_2)^2 + (\sigma_1^2 - \sigma_2^2)^2\right].$$

Therefore, we get:

$$\|H(\boldsymbol{M}_1, \sigma_1^2) - H(\boldsymbol{M}_2, \sigma_2^2)\| \leq \left[\left(\frac{k^2}{\sigma_0^8}\right) + \left(\frac{k^2\varepsilon^2}{\sigma_0^{12}}\right) + \left(\frac{dk^2\varepsilon}{\sigma_0^8}\right) + \left(\frac{k^3\varepsilon^2}{\sigma_0^{10}}\right)\right]\|(\boldsymbol{M}_1, \sigma_1^2) - (\boldsymbol{M}_2, \sigma_2^2)\|$$

$$\equiv \mathrm{poly}\left(k, \varepsilon, d, \frac{1}{\sigma_0}\right)\|(\boldsymbol{M}_1, \sigma_1^2) - (\boldsymbol{M}_2, \sigma_2^2)\|.$$

Finally, since we are summing over $n$ data points, the Hessian Lipschitz constant, is $\mathrm{poly}(n, k, \varepsilon, d, 1/\sigma_0)$.

