# OpenReview forum: "Optimization Can Learn Johnson Lindenstrauss Embeddings"
_NeurIPS.cc/2024/Conference — NeurIPS 2024 poster_

### Official Review · Reviewer_WAK9 · 2024-06-17

**Soundness:** 4
**Presentation:** 4
**Contribution:** 4
**Rating:** 8
**Confidence:** 3

**Summary:**

This work shows that a deterministic optimization procedure can find a matrix $A$ that satisfies the Johnson Lindenstrauss guarantee. That is, a matrix $A$ maps a set of $n$ vectors to a lower dimensional space while preserving all pairwise distances up to some chosen multiplicative distortion. Typically, $A$ is constructed by sampling it from a random matrix distribution with i.i.d. entries. The authors prove that attempting to directly optimize the entries of $A$ through an optimization procedure by minimizing the maximum distortion is prone to being stuck at local minima. However, the authors show that by optimizing the mean of each entry and the entry-wise variance of the distribution $A$ is sampled from, one can maintain a fixed probability of $A$ being a JL-embedding while at the same time guaranteeing the entry-wise variance ‘sigma’ goes to zero. They then show that, when ‘sigma’ is sufficiently small, one may use the optimized expectation of $A$ as the embedding matrix while only slight increasing the maximum distortion, thereby deterministically finding the desired JL embedding matrix $A$. They show that $\rho$-SOSPs (second order stationary points) have sufficiently low variance when $\rho$ is small, and finally show that a method for finding $\rho$-SOSPs suffices to solve the designed optimization problem.

**Strengths:**

Overall, the paper is clearly written and well-motivated. The intuition of the approach and analysis is easy to follow.

The key idea of optimizing the parameters of a random matrix distribution to preserve the JL-embedding property while reducing the entry-wise variance seems like an innovative approach. The authors point out the original space of matrices is contained in this larger probabilistic space, since a deterministic matrix $A$ is equivalent to having mean $A$ and zero variance. Hence, this can be seen as a probabilistic relaxation of the original matrix optimization problem. I have not seen this type of relaxation used in the field of matrix sketching or more generally randomized numerical linear algebra before, and I believe it may be useful for other problems in the area. I am not very familiar with diffusion models, so I cannot speak on the novelty of the approach regarding that area.

The empirical results are also strong in the sense that they show this procedure for constructing a JL embedding tends to achieve a much lower distortion factor than randomized constructions for a fixed dimension.

**Weaknesses:**

The iterative method to find the matrix $A$ takes $\operatorname{poly}(n, k, d)$ steps, i.e., the complexity is proven to be polynomial but not explicitly determined. Since the paper is primarily theoretical with only limited experiments, it is unclear how efficient this method is in practice.

While the results seem very interesting theoretically, the paper could be strengthened by pointing out some practical applications where this improved deterministic JL embedding would be useful. In the applications I am familiar with, oblivious JL embeddings are needed due to the large number of points in the high-dimensional space (e.g., preserving k-means loss). The authors point to embeddings in deep learning as motivation. It is unclear to me as to how the authors expect progress in understanding deterministic JL embeddings to relate to these embeddings in deep learning. Additional clarification of this point would be helpful.

**Questions:**

In the conclusion, you mention the potential for this approach in applications beyond the Johnson Lindenstrauss setting. In your approach for the JL setting, you upper bound the failure probability of the distortion guarantee via the union bound in eqn. (3). This formulation of the objective function seems difficult to translate to other sketching guarantees on $A$ (e.g., projection cost preservation, L2-subspace embedding, affine embedding). Is there any intuitive reason why it may be possible to formulate a relaxed differentiable objective function when the embedding guarantee must hold over an uncountable number of points?

How does learning a JL embedding relate to learning embeddings for application areas discussed in the introduction? In particular, how do you see the results of this paper affecting that line of work? As mentioned above, I think it would be helpful to expand on the link between your result and the motivation of deep learning embeddings given in the intro.

**Limitations:**

Yes

---

> ### Author Rebuttal · Authors · 2024-08-05
>
> Thank you for your detailed and thoughtful review of our paper. We appreciate your recognition of the strengths and innovation of our work. We understand your concerns regarding the practical efficiency of our method and the explicit determination of its complexity. While our paper primarily focuses on the theoretical foundations, we acknowledge the importance of practical applicability. This is why we included an experiment to showcase the superior performance of our method, even with access to only first-order information, demonstrating its computational efficiency.
>
> We understand your questions but first, we want to clarify what we mean by "our approach". Our approach includes lifting the domain to the distributional space, then optimizing the distributions’ parameters and finally obtaining a deterministic solution using variance reduction. We view this as the main contribution of this paper as we provide a different optimization paradigm and show that optimizing indirectly using our framework can be provably better than direct optimization. We expand on this further below.
>
> **Q: [...] Is there any intuitive reason why it may be possible to formulate a relaxed differentiable objective function when the embedding guarantee must hold over an uncountable number of points?**
>
> A: There are many applications of our approach in theoretical results. Often times specialized randomized constructions are used for several tasks (distance preserving in $L_2$ or other norms, $L_0$ sampling, linear-sketching or other tasks as you mentioned). Our hope is to be able to recover these results directly via optimization methods and potentially derandomizing them via variance reduction. While in the case of JL the initial distribution (Gaussian) was very simple, in other applications much more clever constructions are required. Can optimization recover these embeddings in a principled way? While the specific formulation may vary, the underlying principle of optimizing distribution parameters to achieve low variance and deterministic solutions could be adapted to other contexts.
>
>
> **Q: How does learning a JL embedding relate to learning embeddings for application areas discussed in the introduction? In particular, how do you see the results of this paper affecting that line of work? [...].**
>
> A: This is an interesting question. As a first point, we are optimistic that since our contributions include effectively, a new perspective and analysis, it opens the door to studying embeddings of points that have some specific structure. Secondly, in broader terms, deep learning is a general framework applicable to many domains. Frequently in deep learning, we utilize encoder architectures to generate embeddings from data, for which we want certain properties to hold. In this sense, learning a JL embedding can be viewed as a special case of an encoder, where our focus is on preserving the $L_2$ distance. Our work contributes to the deep learning field by informing the design of algorithms of this general principle. Specifically, we show that optimizing directly can lead to “bad” solutions and to overcome this one can choose to optimize in the “richer” space of distributions.
>
> In addition, we believe our results provide a crucial link between deep learning practitioners and theorists who offer theoretical guarantees through classical/randomized algorithms. Embeddings are central to the success of neural networks, driving much of their empirical achievements. However, the lack of provable guarantees for these embeddings remains a significant challenge. By introducing provable guarantees for embeddings via optimization, alongside an algorithmic approach that is still essentially as tractable as a direct optimization approach, our work bridges these two perspectives. It allows us to harness the power of optimization while ensuring that the embeddings maintain desired properties.
>
> This combination not only enhances the reliability of embeddings used in deep learning but also provides insights into the underlying dynamics contributing to the empirical success of deep learning models. This approach can potentially illuminate why certain embeddings work well in practice and how they can be systematically improved with theoretical backing.

---

> > ### Comment · Reviewer_WAK9 · 2024-08-10
> >
> > Thanks to the authors for their reply. I will maintain my score.
> >
> > > Is there any intuitive reason why it may be possible to formulate a relaxed differentiable objective function when the embedding guarantee must hold over an uncountable number of points?
> >
> > I agree that this paper is valuable for raising the possibility of the general approach you've described being used to achieve deterministic optimization based algorithms for these other sketching guarantees. However, doing so indeed seems quite challenging at first glance.
> >
> > > How does learning a JL embedding relate to learning embeddings for application areas discussed in the introduction? In particular, how do you see the results of this paper affecting that line of work?
> >
> > This is a very interesting interpretation of the results in relation to deep learning. Thanks for clarifying.

---

### Official Review · Reviewer_XscG · 2024-07-11

**Soundness:** 2
**Presentation:** 4
**Contribution:** 3
**Rating:** 5
**Confidence:** 3

**Summary:**

The paper proposes to calculate the embedding matrices used in the statement of the Johnson-Lindenstrauss lemma using optimization instead of randomization. The proposed algorithm is a Hessian descent. Authors prove that the algorithm finds the matrix of minimum distortion. Numerical results display the findings

**Strengths:**

JL is a well celebrated result used for proving existence of optimal (low distortion) embeddings. It is stated in the formal result of JL that such embeddings can be found in polynomial time. But we often rely on randomization to exhibit them. It is useful to have an algorithm to calculate the embeddings. The paper tackles a well motivated problem and their presentation is clear and clean.

**Weaknesses:**

The paper lacks complexity analysis of the algorithm. The algorithm proposed requires a full eigenvalue decomposition at every step. It is prohibitive to use this method in any practical scenario. A discussion on the complexity and how to scale the method up (using randomized methods??) would be nice.

**Questions:**

The paper's main claim is that mean distortion embeddings are computationally well studied: spectral methods (SVD / eigenvalue) methods calculate those. Authors claim that the min (instead of mean) distortion method is what they want to find. Can authors explain why the relaxation of f* to f using the probability bound in Eq (2) and invoking union bound to obtain (3) does not reduce the max to a sum. The entire promise of the method is to work with the max directly to minimize distortion. It appears that relaxation of max to a sum using the union bound drops the nice property which was the primary motivation of the work. Can you explain what I am missing of misinterpreting?

**Limitations:**

scaling the method up and a clear argument why relaxation of Eq (3) is not making the problem trivial would impove the paper.

---

> ### Author Rebuttal · Authors · 2024-08-05
>
> Thank you for the review and valuable comments. We appreciate your recognition of the strengths of our work and have addressed your concerns below.
>
> You raise a valid point about the computational expense of second-order methods. We should clarify here that our primary result is that second-order stationary points of the objective function in Equation 4 satisfy the Johnson Lindenstrauss (JL) guarantee. We opted to use a deterministic second-order method in order to achieve an end-to-end derandomized construction of a JL matrix.
>
> Having established these theoretical guarantees, in our empirical evaluations, we do not insist on deterministic computation as our focus is now different, namely to demonstrate that for practical instances that have non-worst-case structure our method significantly outperforms the randomized JL construction. Instead, for simplicity, we use Monte Carlo sampling and randomized first-order methods which are known to converge to second-order stationary points with high probability [Gao et al. 2017]. In practice, second-order methods can be prohibitively expensive but are not necessary for convergence to second-order stationary points if randomness is allowed.
>
> **Q: [...] Can authors explain why the relaxation of f\* to f using the probability bound in Eq (2) and invoking union bound to obtain (3) does not reduce the max to a sum.**
>
> A: The primary objective of our paper is to use optimization-based methods to find matrices that satisfy the JL guarantee, ensuring that the maximum distortion over all points does not exceed a specified threshold $\epsilon$. As we mention, this is a different (and more challenging) objective than what spectral methods like PCA control. Our goal is to show that this can be accomplished deterministically, directly using optimization. We show, however, that if one tries to do this naively, e.g., by directly working in the space of projection matrices, then significant challenges arise (to put it simply, such an approach will not work) because the landscape of the optimization in this space is non-convex, with bad local minima.
>
>
> Therefore, another approach is needed. We do not change the objective -- as you correctly point out, working with the maximum is important in getting the guarantees we wish to give. Instead, we change the optimization approach, and the **space** in which we optimize. To do this, we work in the space of **solution samplers** rather than in the space of projection matrices. So at any current step in the optimization, the current "solution" of the optimizer is not a deterministic projection matrix, but rather a distribution from which one samples such a matrix. For our framework, therefore, we define a new objective $f^*$ (Equation 2) which represents the **probability** of generating a matrix $A$ with maximum distortion exceeding $\epsilon$.  This is the probability that a given solution sampler generates a projection matrix that does not match the JL guarantee. We clarify that *we do not use the union bound to reduce the maximum distortion to the sum of all distortions*. Instead, we reduce $f^*$ to $f$  (Equation 3), i.e. to the sum of probabilities of each point having distortion larger than $\epsilon$ using a randomly generated matrix $A$. To make it more clear, we note that when no randomness is involved, i.e. $\sigma^2=0$, we have that $f^*(M,0) = 0$ means that no projected data point has distortion larger than $\epsilon$ and $f(M,0) = 0$ measures how many projected data points have distortion larger than $\epsilon$. Additionally, it is important to see that, $f^*(M,0) = 0$ if and only if $f(M,0) = 0$.
>
> Ultimately, we show that our process converges to a distribution with no variance, hence, a specific projection matrix that satisfies the JL guarantee.

---

> > ### Comment · Reviewer_XscG · 2024-08-11
> >
> > thanks for the rebuttal. Follow-up questions:
> >
> > * is there a reason why a first-order optimizer would not work and we actually need a second-order (more expensive) optimizer? First order optimizers are not necessarily stochastic / randomized.
> >
> > * If we remove the $\sigma^2$ term from Equation (4), what estimator do we get? is it not SVD?

---

> ### Author Response · Authors · 2024-08-13
>
> Thanks for the response. We answer your follow-up questions below.
>
> **Q: is there a reason why a first-order optimizer would not work and we actually need a second-order (more expensive) optimizer? First order optimizers are not necessarily stochastic / randomized.**
>
> A simple answer to this is that second-order optimization is necessary because, without it, using first-order optimization **without any randomization** results in the optimization getting stuck at the initial point. Generally, to avoid getting stuck throughout the optimization process, you either need second-order information or first-order information combined with randomization. Stochasticity ensures that progress can always be made.
>
> **Q: If we remove the $\sigma^2$ term from Equation (4), what estimator do we get? is it not SVD?**
>
> If we remove the $\sigma^2$ term from Equation (4), we would not obtain SVD. Equation (4) provides a bound on the probability of generating a matrix that achieves **worst-case distortion** larger than a specified threshold $\epsilon$. In contrast, PCA, which relies on SVD, finds an embedding that minimizes **average distortion** of distances between the original and the embedded vectors.
>
> By fixing (or as you mentioned, removing) $\sigma^2$, we would essentially be looking to optimize the mean matrix $M$ for that specific $\sigma^2$. However, completely removing $\sigma^2$ from the optimization process would create an undesirable situation. While we might obtain a “better” mean matrix $M$, we would lose the intended derandomization at the end of the optimization, effectively resulting in $\sigma^2 = 0$.

---

> > ### Comment · Reviewer_XscG · 2024-08-13
> >
> > can you explain this?
> > > second-order optimization is necessary because, without it, using first-order optimization without any randomization results in the optimization getting stuck at the initial point."
> >
> > is this an empirical finding on the numerical experiments you ran? did you try a larger learning rate?
> >
> > or is it theoretically proven and related to the objective function that we are optimizing?
> >
> >
> >
> > > Generally, to avoid getting stuck throughout the optimization process, you either need second-order information or first-order information combined with randomization. Stochasticity ensures that progress can always be made.
> >
> > Can you share a reference or a more formal statement on this?

---

> ### Author Response · Authors · 2024-08-13
>
> **Q: [...] or is it theoretically proven and related to the objective function that we are optimizing?**
>
> A: It is related to the objective function and theoretically proven, so changing the learning rate won’t resolve it. More generally, to avoid getting stuck at any point during the optimization process, we require either a second-order method or a first-order method with some **randomness**. We expand further on this in the next question.
>
> **Q: Can you share a reference or a more formal statement on this?**
>
> A: Our main result is that the second-order stationary points of Equation (4) satisfy the JL guarantee (Theorem 2). To achieve this, we employ a deterministic second-order method, as detailed in our paper, and thus we achieve a **truly derandomized construction** of a JL matrix.
>
> Additionally, it has been shown that first-order methods with randomness also converge to second-order stationary points with high probability, see for example the work of Jin et al. (2017) on saddle points. Thus, whether using a deterministic second-order method or a first-order method with some degree of randomness, we can reach a second-order stationary point, which is precisely what is required to ensure the JL guarantee.
>
> We hope this clarifies your questions.

---

### Official Review · Reviewer_y6Yv · 2024-07-11

**Soundness:** 2
**Presentation:** 3
**Contribution:** 3
**Rating:** 5
**Confidence:** 3

**Summary:**

The paper considers using the optimization method to "learn" the Johnson Lindenstrauss transform. The paper first shows that the naive objective may not be good enough -- there are stationary points that are sub-optimal. Instead, they consider the way that optimize the random Gaussian space rather than the concrete matrix. Then the authors give an optimization method and show that using this way, every stationary point is a good point and claim that this gives way to deterministic learns the JL matrix. Finally, the paper gives an experiment that shows the advantages of the proposed method.

**Strengths:**

The theoretical analysis of this paper is very interesting. To my knowledge, there are very few results about analyzing the landscape of the learned sketching matrix and this paper gives a strong analysis. The experiments also show the advantages of the proposed method. The presentation of the paper is also good.

**Weaknesses:**

I am still confused about some parts of the paper. I can raise the score if the authors can adjust this. (see the below question)

**Questions:**

1. I am still a little confused about the main conclusion of the results of this paper. That is -- it gives a deterministic construction of the JL lemma, or it gives a better optimization way and it works well empirically? (as the author mentioned, the bound of the JL lemma can not be improved)

2. The equation (4) is about probability, and B.1 says they use Monte Carlo sampling, however, would it means that the proposed method still contains some randomness part?

3. In the experiment section, the paper compares the proposed method with the JL lemma. It will make this stronger if the comparison with equation (1) is also given.

**Limitations:**

See above questions.

---

> ### Author Rebuttal · Authors · 2024-08-05
>
> Thank you for your constructive comments and for appreciating our analysis. We agree with you that there are not many results analyzing the landscape of the learned sketching matrix. We address your questions below.
>
> **Q: [...] it gives a deterministic construction of the JL lemma, or it gives a better optimization way and it works well empirically? [...].**
>
> A: The answer is both. Our main contribution is that one can use direct optimization to efficiently find "good" embeddings. We propose a novel optimization objective and analyze its performance from two distinct angles:
>
> - First, we take a worst-case perspective where we theoretically show that optimization can obtain matrices that are always at least as good as the guarantees of the Johnson Lindenstrauss (JL) lemma. This establishes that our method is worst-case optimal, given that the JL lemma guarantees cannot be improved.
> - Secondly, we demonstrate that for practical instances that have non-worst-case structure our method significantly outperforms the randomized JL construction. This is intuitively right as the randomized construction is data agnostic.
>
> Along the way, we establish that naive methods do not have these guarantees as the space of projection matrices is non-convex and has bad local minima. Instead, we need to work in a slightly larger space of samplers, and there, our optimization method will find a deterministic solution.
>
> In summary, our theoretical guarantees state that we can achieve the JL guarantee in the worst case, but we showcase that in practice we can go beyond that, if the data allows it.
>
>
> **Q: The equation (4) is about probability, and B.1 says they use Monte Carlo sampling, however, would it means that the proposed method still contains some randomness part?**
>
> A: As mentioned above our main contribution is the analysis of this novel objective. While this objective (Equation 4) involves probabilities, they have a closed form expression and can be calculated exactly without any sampling or randomness. Our main result is that second-order stationary points of this objective function satisfy the JL guarantee. Using a deterministic second-order method, we achieve a **truly derandomized construction** of a JL matrix.
>
> Having established these theoretical guarantees, in our empirical evaluations, we do not insist on deterministic computation as our focus is now different (as explained in the previous question). Instead, for simplicity, we use Monte Carlo sampling and randomized first-order methods which are known to converge to second-order stationary points with high probability [Gao et al. 2017]. In practice, second-order methods can be prohibitively expensive but are not necessary for convergence to second-order stationary points if randomness is allowed.
>
>
>
> **Q: In the experiment section, the paper compares the proposed method with the JL lemma. It will make this stronger if the comparison with equation (1) is also given.**
>
> A: Thank you for the suggestion. However, as we show in Theorem 1, optimizing Equation 1 directly in the matrix space can lead to bad solutions with high distortion, whereas our method is guaranteed to succeed. Since we showed that taking the approach of Equation 1 can (provably) fail, we decided that it would not provide a fair comparison and would distract from the comparison with the randomized JL construction.

---

> > ### Comment · Reviewer_y6Yv · 2024-08-12
> >
> > Thanks for the response.
> >
> > - Derandomization: Thanks for pointing this out. However, the calculation of the closed form of expression (4) seems to be non-trivial and some discussion about the computation should be included to support the claim (e.g., the time complexity). This is the main reason I keep my score.
> >
> > - Experiment: I agree that "optimizing Equation 1 directly in the matrix space can lead to bad solutions with high distortion" However, it was shown in the specific hard instance and the comparison over the real-world datasets will still be interesting.

---

> > > ### Author Response · Authors · 2024-08-13
> > >
> > > Thank you for your response.
> > >
> > > Regarding the first point, it's important to note that the calculations of the gradient involve **only** cumulative distribution functions of known distributions, i.e. the chi-squared distribution, which simplifies the computations. We acknowledge that this can be explained more clearly and we will incorporate a discussion on the computational aspects in our work.
> > >
> > > We appreciate all your valuable feedback and will incorporate it to strengthen our paper. Thank you for your insightful comments and suggestions.

---

### Official Review · Reviewer_pTrj · 2024-07-19

**Soundness:** 4
**Presentation:** 3
**Contribution:** 4
**Rating:** 7
**Confidence:** 3

**Summary:**

This paper investigates the problem of using optimization-based approaches to learn Johnson Lindenstrauss(JL) embedding. The authors proposed a new framework to achieve the JL guarantee via optimization, instead of the traditional randomized methods. Similar with diffusion models, the authors proposed a novel method that uses optimization in an extended space of random Gaussian solution samplers, which circumvents direct optimization in non-convex landscape. Their approach uses second-order descent, gradually reduces the variance without increasing the expected distortion of the sampler, then can identify a specific projection matrix with the Gaussian distribution. Overall, theoretical guarantees and empirical results demonstrate that this method efficiently achieves a deterministic solution that satisfies the JL guarantee.

**Strengths:**

The paper is well-written. The state of the art is well discussed by an extensive literature review. The proposed method combining optimized-based approaches and Johnson Lindenstrauss embeddings is an innovative contribution to the field.

The paper is technically sound, provides rigorous theoretical analysis and proofs.

**Weaknesses:**

It would be helpful to understand the main results if section 4 could be more organized, such as using subsections.

**Questions:**

1. Could you explain the statement in lines 215-216? Are the values of 1/(3n) and 1/3 derived based on the chosen value of $\epsilon$?

2. Line 336, there is a typo  “appplicability”.

3. Regarding the notation $\rho$-second-order stationary points($\rho$-SOSP), the paper uses $\rho$-second-order stationary points in some sections and uses $\rho$-SOSP in others.

**Limitations:**

No potential negative societal impact.

---

> ### Author Rebuttal · Authors · 2024-08-05
>
> Thank you for your thoughtful feedback and for recognizing the innovation in our work.
>
> **Q: Could you explain the statement in lines 215-216? Are the values of 1/(3n) and 1/3 derived based on the chosen value of** $\epsilon$**?**
>
> A: Yes, that's exactly correct: you can choose $\epsilon$ appropriately to get a $1/(3n)$ probability for any data point to violate the threshold constraint. Thus, if you sum these $n$ probabilities you get $1/3$.
>
> Thank you for your keen observations on the paper's typo and notation inconsistencies, we fixed it and made it consistent throughout.

---

> > ### Comment · Reviewer_pTrj · 2024-08-13
> >
> > Thanks for the response. I will maintain my score.

---

### Author Rebuttal · Authors · 2024-08-06

We thank the reviewers for their insightful and constructive feedback on our submission. We appreciate the time and effort they have dedicated to reviewing our work. We are encouraged by their positive reception, noting that they found our contribution innovative (pTrj, WAK9), our analysis strong (y6Yv), and the problem well-motivated (XscG).

Furthermore, reviewer WAK9 noted that our probabilistic relaxation can be useful for other problems in matrix sketching and randomized numerical linear algebra. Reviewers y6Yv and WAK9 also found our experiments valuable, highlighting that they effectively demonstrate the advantages of our method.

We address the reviewers' comments below and will incorporate all feedback.

---

### Decision · Program_Chairs · 2024-09-25

**Decision:**

Accept (poster)

**Comment:**

This paper proposes a data-driven optimization approach for embedding that is in stark contrast to randomized methods such as the Johnson-Lindenstraus (JL). While solving the distance-preserving objective is intractable over the projection matrices, this paper considers an optimization program that searches a solution on a larger space and shows that the algorithm converges in polynomial steps.

The idea that optimizing over an expanded and carefully constructed space is novel. The results are sound, in the sense that the solution returned by the optimization program satisfies the JL guarantee; this is the first result of its kind which does not rely on random projection matrix. Empirically, it appears that the optimization-based approach enjoys lower distortion. That said, there are also weaknesses regarding the computational cost and empirical viability of the method as it relies on second order optimization.

Overall, the techniques presented in this paper may inspire a new line of research and represent a nice addition to NeurIPS program.